# Train Faster, Perform Better: Modular Adaptive Training in Over-Parameterized Models

**Yubin Shi**[1]    **Yixuan Chen**[1]    **Mingzhi Dong**[1,*]    **Xiaochen Yang**[2,*]
**Dongsheng Li**[3]    **Yujiang Wang**[4]    **Robert Dick**[5]    **Qin Lv**[6]
**Yingying Zhao**[1]    **Fan Yang**[7]    **Tun Lu**[1]    **Ning Gu**[1]    **Li Shang**[1,*]

[1]China and Shanghai Key Laboratory of Data Science, School of Computer Science, Fudan University
[2]School of Mathematics Statistics, University of Glasgow
[3]Microsoft Research Asia, Shanghai, China
[4]Department of Engineering Science, University of Oxford
[5]Department of Electrical Engineering and Computer Science, University of Michigan
[6]Department of Computer Science, University of Colorado Boulder
[7]School of Microelectronics, Fudan University

## Abstract

Despite their prevalence in deep-learning communities, over-parameterized models convey high demands of computational costs for proper training. This work studies the fine-grained, modular-level learning dynamics of over-parameterized models to attain a more efficient and fruitful training strategy. Empirical evidence reveals that when scaling down into network modules, such as heads in self-attention models, we can observe varying learning patterns implicitly associated with each module's trainability. To describe such modular-level learning capabilities, we introduce a novel concept dubbed modular neural tangent kernel (mNTK), and we demonstrate that the quality of a module's learning is tightly associated with its mNTK's principal eigenvalue $\lambda_{\max}$. A large $\lambda_{\max}$ indicates that the module learns features with better convergence, while those miniature ones may impact generalization negatively. Inspired by the discovery, we propose a novel training strategy termed Modular Adaptive Training (MAT) to update those modules with their $\lambda_{\max}$ exceeding a dynamic threshold selectively, concentrating the model on learning common features and ignoring those superfluous ones. Unlike most existing training schemes with a complete BP cycle across all network modules, MAT can significantly save computations by its partially-updating strategy and can further improve performance. Experiments show that MAT nearly halves the computational cost of model training and outperforms the accuracy of baselines.

## 1  Introduction

The proliferation of large-scale pre-trained models (Devlin et al., 2018; Brown et al., 2020; Dosovitskiy et al., 2020) demonstrates the advantages of over-parameterized models. Those models can adequately fit the training data with their sizeable parameters far exceeding the data scale (Du et al., 2018), and such over-parameterizations are known to be essential in model optimization and generalization and are also crucial to their success in deep learning (Liu et al., 2022). However, challenges remain. Over-parameterized models are expensive to train; for example, training large language models (LLMs) such as GPT-3 (Brown et al., 2020; Zaheer et al., 2020) may take weeks to months, even on a large collection of powerful GPUs.

---

[*]Corresponding authors.

37th Conference on Neural Information Processing Systems (NeurIPS 2023).

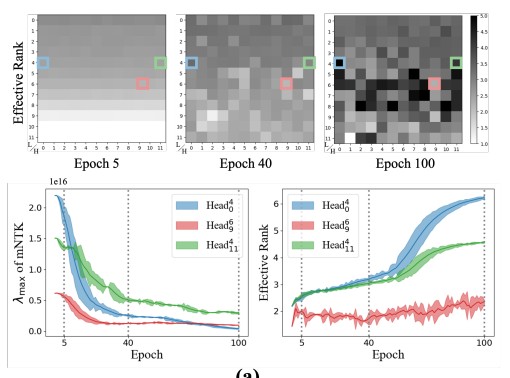

**Modular Adaptive Training (MAT)**

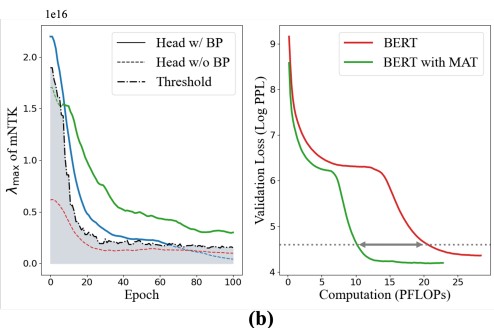

(a)                 (b)

Figure 1: Characteristics of training BERT on WikiText-2. Figure (a) demonstrates the joint variation of effective rank and $\lambda_{\max}$ across the attention heads. $\text{Head}_i^l$ refers to the $i^{\text{th}}$ attention head in the $l^{\text{th}}$ layer. Figure (b-left) illustrates the idea of MAT, which governs the heads training by a dynamic threshold. Using MAT speeds up convergence and achieves lower validation loss (b-right).

Generally, most over-parameterized models consist of various functioning modules connected layer-wisely, e.g., heads in self-attention models like Transformer (Vaswani et al., 2017), experts in mixture-of-experts (MoE) blocks (Shazeer et al., 2017b), or filters in convolutional neural networks (CNN). This module-based architecture inevitably leads to a question: *Has the computational resource been spent efficiently and effectively across modules during model optimization?* To this end, we dive into the modular-level training behaviors of over-parameterized models and pursue a more efficient and fruitful training strategy. We propose modular Neural Tangent Kernel (mNTK), which is derived from the vanilla NTK (Jacot et al., 2018; Fort et al., 2020), as a powerful tool for understanding the fine-grained learning dynamics of each network module. We compute mNTK as the first-order approximation of a network module's evolution during training, and it reflects the gradient correlation among the training data module-wisely. The eigenspectrum of mNTK describes the learning dynamics of a module, and its principal eigenvalue $\lambda_{\max}$ indicates the degree of consistency in the gradient direction for learning the most common features in data.

To illustrate the module-level training dynamics intuitively, we employ the standard BERT (Devlin et al., 2018) as an example and consider each attention head as a specific module. In addition to $\lambda_{\max}$, we adopt the effective rank (Roy & Vetterli, 2007) to measure the effective dimensionality of a feature matrix, like an attention matrix, and those metrics are computed at the end of each training epoch. Figure 1(a) presents the trend of training dynamics of BERT, and we can observe asynchronous learning behaviors across modules and training iterations. We explain those observations using $\lambda_{\max}$ as an indicator of trainability to measure how effectively gradient descent can optimize network parameters. As discussed in Bowman & Montufar (2022), the network is biased to learn the top eigenfunctions of the NTK over the entire input space, while the parameters with very low eigenvalues barely update. In Figure 1(a), the attention head in blue marker has large $\lambda_{\max}$, meaning that it has large consistency over data samples on feature learning and is thus apt to converge with better trainability. Indeed, these heads contain richer information as reflected through effective rank. In contrast, $\lambda_{\max}$ of the red attention head oscillates at a low level, likely due to learning diverse features or even noise that may be hard to converge, and thus updating it brings little benefit to training loss.

Our analysis (Section 2) on trainability and generalization demonstrates that over-parameterized models exhibit modularly and temporally varying asynchronous learning behaviors. *Modules with large mNTK principal eigenvalues learn features that are more rapidly learned, and those miniature ones with limited impact on training loss negatively influence generalization. Such asynchronous modularly and temporally training dynamics result in inefficient resource utilization by existing training schemes.* Motivated by these analyses, we propose a simple yet effective training method termed Modular Adaptive Training (MAT), aiming at dynamically training partial modules to improve training efficiency. As shown in Figure 1(b), we design a dynamic threshold (the dash-dotted line), according to modular difference and temporal variation of mNTK $\lambda_{\max}$. MAT only back-propagates the parameters of those modules whose $\lambda_{\max}$ is larger than the threshold. This forces the network to

concentrate on learning the common features and ignore the inconsistent ones, preventing it from fitting superfluous features or noises. This work makes the following contributions.

- We empirically and theoretically reveal the associations between modular-level training dynamics and the proposed modular Neural Tangent Kernel (mNTK).
- We propose Modular Adaptive Training (MAT) with a dynamic threshold to selectively update modules during back-propagation to improve learning efficiency.
- Experimental results verify that MAT can significantly reduce the training computation, improve performance, and generalize well to different over-parameterized models.

## 2 Analysis of Modular Neural Tangent Kernel in Over-Parameterized Models

In this section, we first introduce the definition of Neural Tangent Kernel (NTK) and modular Neural Tangent Kernel (mNTK). Next, we present empirical analysis revealing the eigenspectrum characteristics of the mNTK, including the imbalanced distribution of mNTK eigenvalues and the modular and temporal diversity of mNTK eigenvalues during training. Finally, we provide theoretical analysis to show that trainability of over-parameterized structured models is closely related to the largest mNTK eigenvalues; on the other hand, learning features associated with the miniature eigenvalues have negative impact on generalization.

### 2.1 Modular Neural Tangent Kernel (mNTK)

Let $\mathcal{X}$ denote the set of $n$ training instance, which are i.i.d. drawn from an unknown distribution $\mathcal{D}$, and let $\mathcal{Y} \in \mathbb{R}^{nk}$ denotes the targets. We study the network function $f$ parameterized by $\boldsymbol{\theta}$ aiming to map the input vector $\mathcal{X}$ to output vector $\mathcal{Z}$, termed as $\mathcal{Z} = f(\mathcal{X}; \boldsymbol{\theta})$ where $\boldsymbol{\theta} \in \mathbb{R}^m$ and $\mathcal{Z} \in \mathbb{R}^{nk}$. Following the original definition of Neural Tangent Kernel (NTK) (Jacot et al., 2018; Wang et al., 2020), we introduce modular NTK for fine-grained analysis of structured deep networks as below.

**Definition 1** *(Modular Neural Tangent Kernel (mNTK)). Suppose model $f$ contains $L$ disjoint modules $\boldsymbol{\theta} = \{\boldsymbol{\theta}^1, \boldsymbol{\theta}^2, \ldots, \boldsymbol{\theta}^L\}$ where $\boldsymbol{\theta}^l$ denote the parameters of $l^{th}$ module. We define mNTK as a matrix by $\boldsymbol{\Theta}^l(\mathcal{X}, \mathcal{X}) = J_{\boldsymbol{\theta}^l}(\mathcal{X}) J_{\boldsymbol{\theta}^l}(\mathcal{X})^\top$, where $J_{\boldsymbol{\theta}^l} = \nabla_{\boldsymbol{\theta}^l} f(\mathcal{X}; \boldsymbol{\theta}^l)$ denotes the Jacobian of the function $f$ at the points $\mathcal{X}$ with respect to the $l^{th}$ module's parameters $\boldsymbol{\theta}^l$.*

$\boldsymbol{\Theta}^l$ is a positive semi-definite real symmetric matrix and can be eigen-decomposed as $\boldsymbol{\Theta}^l = \mathbf{U}^l \boldsymbol{\Lambda}^l \mathbf{U}^{l^\top} = \sum_{i=1}^{nk} \lambda_i^l \mathbf{u}_i^l \mathbf{u}_i^{l^\top}$ with $nk$ non-negative eigenvalues $\boldsymbol{\lambda}(\boldsymbol{\Theta}^l) = \{\lambda_1^l, \lambda_2^l, ..., \lambda_{nk}^l\}$. Note that a network can be partitioned into modules in flexible ways, for example, by dividing layers within a deep network or components of a network with inherent structures (e.g., attention heads in Transformer or convolutional filters in a CNN). Since each module has its own mNTK, the integral NTK of a model can be computed as the sum of mNTKs of each module (Yang & Littwin, 2021):

$$\boldsymbol{\Theta}(\mathcal{X}, \mathcal{X}) \stackrel{(i)}{=} \sum_{p=1}^{m} J_{\boldsymbol{\theta}_p}(\mathcal{X}) J_{\boldsymbol{\theta}_p}(\mathcal{X})^\top \stackrel{(ii)}{=} \sum_{l=1}^{L} \sum_{\boldsymbol{\theta}_p \in \boldsymbol{\theta}^l} J_{\boldsymbol{\theta}^l}(\mathcal{X}) J_{\boldsymbol{\theta}^l}(\mathcal{X})^\top \stackrel{(iii)}{=} \sum_{l=1}^{L} \boldsymbol{\Theta}^l(\mathcal{X}, \mathcal{X}), \qquad (1)$$

where $(i)$ decomposes the matrix multiplication into the sum of vector multiplication; $(ii)$ gathers addends by each module; $(iii)$ follows the definition of mNTK.

### 2.2 Empirical Analysis

In this subsection, we apply BERT to WikiText-2 for language modeling and regard each layer as a module to analyze the properties of eigenvalue distribution of all mNTKs, as well as the modular and temporal variation of eigenvalues during the training process.

**The principal eigenvalue of mNTK dominates the eigenspectrum of mNTK.** Figure 2(a) shows the first 16 eigenvalues of all layer-wise mNTKs at the $10^{th}$ epoch. Apparently, the eigenvalue distribution of each layer-wise mNTK is imbalanced where there exists a single large eigenvalue $\lambda_1^l$ ($\lambda_1$ will be denoted by $\lambda_{\max}$ hereafter) and a large number of small eigenvalues. In particular, the largest eigenvalue often has 2 to 4 orders of magnitude larger than that of the rest eigenvalues. These findings indicate that the principal eigenvalue of mNTK dominates the spectrum of the mNTK.

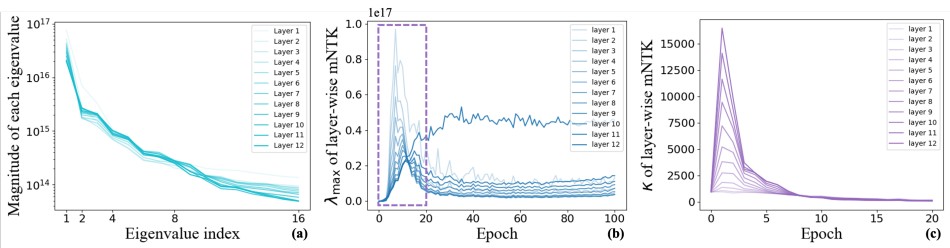

Figure 2: Training dynamics characterized by layer-wise mNTK of BERT trained on WikiText-2. (a): The first 16 eigenvalues distribution of layer-wise mNTKs at the $10^{\text{th}}$ epoch. (b): Variation of $\lambda_{\max}$ of layer-wise mNTKs. (c): Variation of $\kappa$ of layer-wise mNTKs during the first 20 epochs.

The phenomenon is also indicated by Xiao et al. (2020) in terms of NTK. Furthermore, according to the connection between NTK and mNTK (Eq. 1), the spectrum of NTK is dominated by the principal eigenvalues of all mNTKs. Below, we utilize $\lambda_{\max}$ for fine-grained optimization analysis of over-parameterized model.

**The variation of principal eigenvalues of mNTKs is highly asynchronous across modules.**
Figure 2(b) shows the variation of $\lambda_{\max}$ of all layer-wise mNTKs during the training process. It is worth noting that the principal eigenvalues of mNTKs exhibit different trends during the training process. Intuitively speaking, NTK (also mNTK) captures the training dynamics throughout the training process, and its principal eigenvector corresponds to the direction of the steepest descent in loss. Specifically, for mNTKs of shallow layers, the magnitude of $\lambda_{\max}$ is large after model initialization, and it decreases and converges fast in the early training stages. A potential explanation is that shallow layers learn simple and common features of data samples (Yosinski et al., 2014; Zhang et al., 2019), which contribute significantly to model convergence in the early training stages. In contrast, for mNTK of the deep layer, the magnitude of $\lambda_{\max}$ is small after initialization and gradually increases during training. The reason is that deep layers learn complex and unique features, relying on the stability of simple features in shallow layers, which occurs in the later training stages.

To analyze the training dynamics, condition number is also commonly used (Xu et al., 2021; Xiao et al., 2020), defined as $\kappa = \lambda_{\max}/\lambda_{\min}$. Figure 2(c) shows the variation of $\kappa$ for different modules during training. We observe that the layer-wise condition number exhibits asynchronization among different modules only in the early training stages (especially the first 10 epochs) and $\kappa$ of all modules quickly converges to the same value. This reveals that the condition number is hard to provide sustained information for fine-grained optimization analysis, unlike the principal eigenvalue which is capable of capturing the effective information throughout the training process.

**The temporal variation of the principal eigenvalue of mNTK indicates the generalization ability.**
In addition to the above analysis depicting the asynchronization between different modules, we examine the temporal variation of the principal eigenvalue given a specific module and establish its connection with the generalization ability. In detail, we train the same model using masked modeling on half sequence length, which makes the model prone to overfitting. As shown in Figure 3(a), the validation loss starts to increase from the $45^{\text{th}}$ epoch while the training loss continually decreases. Figure 3(b-e) present the temporal variation of the principal eigenvalue for the $3^{\text{rd}}$, $6^{\text{th}}$, $9^{\text{th}}$, and $12^{\text{th}}$ layers of the model during the training process, respectively. During the first $45^{\text{th}}$ epochs, i.e., the non-overfitting stage, the principal eigenvalues of these layers continually decrease. However, after the $45^{\text{th}}$ epoch when the model enters the overfitting stage, the principal eigenvalues converge to excessively low values. A possible reason for overfitting is that the optimization direction of the model starts to align with the direction of small eigenvalues, which means that the model is learning superfluous features that are too specific or even noisy.

## 2.3 Analysis of Trainability and Generalization

Using the tool of mNTK, we show that the optimization properties of over-parameterized structured models are closely related to eigenvalues of mNTKs.

Suppose for any module $\boldsymbol{\theta}^l$, $\lambda_{\min}(\boldsymbol{\Theta}^l) = \lambda_{nk}^l > 0$. Let $\mathcal{L}$ denote the loss function and $\nabla_{\boldsymbol{\theta}} \mathcal{L}(\boldsymbol{\theta})$ the gradient w.r.t parameter $\boldsymbol{\theta}$. For a step of gradient descent, loss reduction can be characterized by the

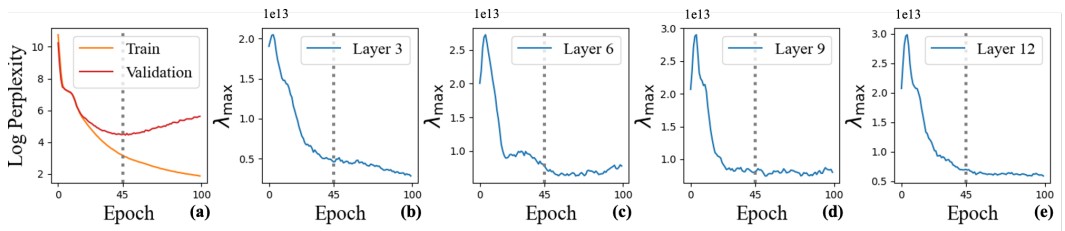

Figure 3: Training dynamics in the overfitting Case of 4-layer BERT trained by 64 token MLM task.

directional derivative of the loss function (Wang et al., 2020):

$$
\begin{aligned}
\Delta \mathcal{L} &\overset{(i)}{=} \lim_{\epsilon \to 0} \frac{\mathcal{L}(\boldsymbol{\theta} + \epsilon \nabla_{\boldsymbol{\theta}} \mathcal{L}(\boldsymbol{\theta})) - \mathcal{L}(\boldsymbol{\theta})}{\epsilon} \overset{(ii)}{\approx} \nabla_{\boldsymbol{\theta}} \mathcal{L}(\boldsymbol{\theta})^{\top} \nabla_{\boldsymbol{\theta}} \mathcal{L}(\boldsymbol{\theta}) \\
&\overset{(iii)}{=} \nabla_{\mathcal{Z}} \mathcal{L}(\boldsymbol{\theta})^{\top} (\nabla_{\boldsymbol{\theta}} f(\boldsymbol{\theta})^{\top} \nabla_{\boldsymbol{\theta}} f(\boldsymbol{\theta})) \nabla_{\mathcal{Z}} \mathcal{L}(\boldsymbol{\theta}) \\
&\overset{(iv)}{=} \nabla_{\mathcal{Z}} \mathcal{L}(\boldsymbol{\theta})^{\top} (\sum_{l=1}^{L} \boldsymbol{\Theta}^{l}) \nabla_{\mathcal{Z}} \mathcal{L}(\boldsymbol{\theta}) \overset{(v)}{=} \sum_{l=1}^{L} \sum_{i=1}^{nk} \lambda_{i}^{l} \left( \mathbf{u}_{i}^{l \top} \mathcal{Y} \right)^{2},
\end{aligned}
\tag{2}
$$

where $(i)$ follows the definition of directional derivative; $(ii)$ follows the first-order Taylor expansion; $(iii)$ follows chain rule of derivative; $(iv)$ follows Eq. 1; and $(v)$ follows the eigen-decomposition of mNTK under the assumption of squared error loss (Arora et al., 2019). Following the work of Arora et al. (2019), we assume that true labels align well with top eigenvectors, i.e., $(\mathbf{u}_{i}^{l \top} \mathcal{Y})^{2}$ is large for large $\lambda_{i}^{l}$. Thus, the directional derivative of the loss function can be regarded as closely related to the eigenspectrum of mNTKs.

(1) The relation between **trainability** and principal eigenvalues of mNTKs.

Based on the first observation of empirical analysis, i.e., the spectrum of each mNTK is dominated by the principal eigenvalues, we utilize $\lambda_{\max}$ of mNTKs as the nearly equivalent proxy of the spectrum of mNTKs. Therefore, Eq. 2 is simplified as:

$$
\Delta \mathcal{L} \approx \sum_{l=1}^{L} \lambda_{\max}^{l} \left( \mathbf{u}_{1}^{l \top} \mathcal{Y} \right)^{2}.
\tag{3}
$$

The above equation suggests that the loss decreases faster along the eigenspaces that correspond to larger eigenvalues. Given the fact that the principal eigenvalues of mNTKs are highly asynchronous across modules, we suggest selectively training the modules that are active with larger $\lambda_{\max}^{l}$ to achieve efficient learning with limited computational resources.

(2) The relation between **generalization ability** and eigenvalues of mNTKs.

Figure 4 shows the eigen-spectrum distribution of a 4-layer BERT model at the $10^{\text{th}}$ epoch, by regarding each head as a module and calculating 256 eigenvalues for each head. Among all $4 \times 12 \times 256$ eigenvalues, the distribution of eigen-spectrum exhibits two distinct regions: the left region with a large number of small eigenvalues and the right region with a small number of large eigenvalues. The two regions are widely separated. According to Eq. 3, the eigenspaces corresponding to miniature eigenvalues in the left region have negligible impact on training loss minimization, while those in the right region make dominant contributions. Following the definition of Oymak et al. (2019), we refer

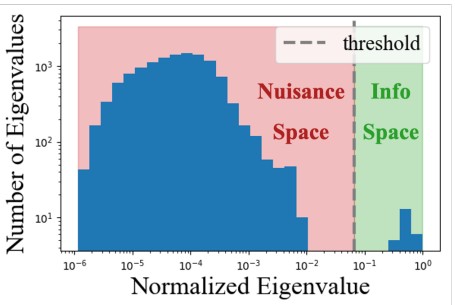

Figure 4: The normalized eigen-spectrum distribution exhibits two distinct regions, termed information space and nuisance space.

to the right region as the *information space* where the training error decreases rapidly, and the left region as the *nuisance space* where training is slow and may cause overfitting.

Model training with more parameter updates tends to have longer distance to the initializations $\boldsymbol{\theta}_0$ (Hoffer et al., 2017). Based on Lemma 5.4 in Arora et al. (2019), longer distance results in relatively larger Rademacher complexity, as $\mathcal{R} \propto \|\boldsymbol{\theta} - \boldsymbol{\theta}_0\|_F$, and larger $\mathcal{R}$ results in relatively worse generalization. As a consequence, we suggest selectively stopping the training in some modules so as to stop the increase of the distance to the initializations in these directions. Specifically, we propose to stop training the modules with principal eigenvalues falling into the nuisance space so as to enjoy better generalization and efficient learning simultaneously.

Let's consider the inter-module training dynamics on model generalization. During the training process, given modules with variant principal eigenvalues, it often occurs that even though a small number of modules still lie in the information space, the most of others already fall into the nuisance space. The modules located in the information space, i.e., those with principal eigenvalues exceeding the threshold, are trained to learn informative features that facilitate loss minimization. Conversely, the modules falling into the nuisance space, i.e., those with principal eigenvalues below the threshold, are prone to learn superfluous or even noise features, which significantly increase the Rademacher complexity, and deteriorate their generalization ability. *This analysis indicates that the existing training scheme overlooks the asynchronization among modules, resulting in inefficient utilization of the computational resource for loss minimization during model training.*

## 3   Modular Adaptive Training

Based on the analysis in the previous section, we advocate dynamically optimizing partial module parameters in back propagation, termed as Modular Adaptive Training (MAT). The forward propagation is computed as usual; during back propagation, MAT quantifies module parameters in terms of **modular policy** and **temporal policy** such that only subsets of modules are updated. Below, we present the proposed modular policy and temporal policy. Algorithm 1 summarizes the proposed MAT algorithm.

**Modular Policy.** Following the analysis that trainability and generalization of an over-parameterized model are related to the principal eigenvalues of module parameters, we propose to only update partial module parameters with $\lambda_{\max}$ greater than a certain threshold during the model training; this policy is termed as modular policy.

Specifically, suppose the model parameters consist of $L$ module parameters and each mNTK has $nk$ positive eigenvalues, and let $\lambda_i^l$ denote $i^{\text{th}}$ eigenvalue of the mNTK computed over the parameters $\boldsymbol{\theta}^l$. We define $\widetilde{\boldsymbol{\Lambda}} = \{\lambda_i^l\}_{i,l=1}^{nk,L}$ as the set of eigenvalues calculated over all mNTKs, and $\widetilde{\lambda}_{\max}, \widetilde{\lambda}_{\min}$ as the maximum and minimum value of the set $\widetilde{\boldsymbol{\Lambda}}$, respectively. To select the appropriate modules, we introduce a hyperparameter ratio $\alpha \in (0,1)$, thus the corresponding eigenvalue threshold $\lambda_\alpha$ is computed as:

$$\lambda_\alpha = \widetilde{\lambda}_{\min} + (\widetilde{\lambda}_{\max} - \widetilde{\lambda}_{\min}) \times \alpha. \tag{4}$$

According to the eigenvalue threshold $\lambda_\alpha$, all modules are split into *information modules* and *nuisance modules* as:

$$\mathcal{I}_t = \{\boldsymbol{\theta}^l | \lambda_{\max}(\boldsymbol{\Theta}_t^l) \geq \lambda_\alpha\}, \mathcal{N}_t = \{\boldsymbol{\theta}^l | \lambda_{\max}(\boldsymbol{\Theta}_t^l) < \lambda_\alpha\}. \tag{5}$$

As a result, the computation cost of back propagation is reduced.

We empirically observe that $\alpha \in (0,1)$ is relatively stable over the training process for a certain model and dataset. Therefore, $\alpha$ can be empirically assigned at the early stage of training after warming up, or automatically learned by a meta network.

**Temporal Policy.** The previous empirical analysis shows that during training, after the eigenvalues of a module's mNTK enter the nuisance region, further training this module can lead to poor generalization. Therefore, to avoid overfitting, we terminate the training of this module according to the temporal variation of the principal eigenvalue; we call this the temporal policy.

For each module $\boldsymbol{\theta}^l$, its principal eigenvalue after warming up is recorded as the initial value $\lambda_{\max}(\boldsymbol{\Theta}_0^l)$. During each training episode $t$, the temporal variation of principal eigenvalue is defined as $\Delta_t^l = |\lambda_{\max}(\boldsymbol{\Theta}_t^l) - \lambda_{\max}(\boldsymbol{\Theta}_0^l)|$. The information modules will be early stopped if the difference in the temporal variation between two subsequent episodes relative to the initial variation is smaller

than a pre-defined threshold $\beta$:

$$\frac{|\Delta_t^l - \Delta_{t-1}^l|}{\Delta_1^l} < \beta. \tag{6}$$

$\beta$ is empirically bounded by eigenvalue variation within the nuisance region.

## 4 Related Work

**Optimization Analysis using NTK.** Neural Tangent Kernel (NTK) (Jacot et al., 2018), which calculates the Gram matrix of Jacobian, is known as a powerful tool to analyze convergence and generalization properties (Arora et al., 2019). Modern neural networks are generally over-parameterized with positive definite NTKs, meaning that theoretically the gradient flow always converges to the equilibrium where the training loss is zero (Chizat & Bach, 2018; Bombari et al., 2022). A line of works study the training dynamics of over-parameterized models (Li et al., 2020; Liu et al., 2022) based on NTK, mainly under the assumption of two-layer infinite-width networks. Many papers (Xiao et al., 2020) study the spectrum of the NTK, and find in particular the largest eigenvalue dominates the training regime (Jacot et al.,

---

**Algorithm 1** Modular Adaptive Training

**Require:** training data $\{\mathbf{x}_i, \mathbf{y}_i\}_{i=1}^n$, sample number $S$, modules set $\mathcal{M}$, modular threshold $\alpha$, temporal threshold $\beta$.
1: $\mathcal{I}, \mathcal{N} = \mathcal{M}, \varnothing$
2: **while** not converged and $|\mathcal{I}| > 0$ **do**
3:     Compute $\nabla_{\boldsymbol{\theta}} f(\mathbf{x}; \boldsymbol{\theta})$ for samples $\{\mathbf{x}_i, \mathbf{y}_i\}_{i=1}^S$
4:     $\widetilde{\boldsymbol{\Lambda}} = \varnothing$
5:     **for** $\boldsymbol{\theta}^l$ in $\mathcal{M}$ **do**
6:         $\boldsymbol{\Theta}^l = \nabla_{\boldsymbol{\theta}^l} f(\mathbf{x}; \boldsymbol{\theta}) \nabla_{\boldsymbol{\theta}^l} f(\mathbf{x}; \boldsymbol{\theta})^\top$
7:         Compute eigen-decomposition $\boldsymbol{\Theta}^l = \mathbf{U}^l \boldsymbol{\Lambda}^l \mathbf{U}^{l\top}$
8:         $\Delta_t^l = |\lambda_{\max}(\boldsymbol{\Theta}^l) - \lambda_{\max}(\boldsymbol{\Theta}_0^l)|$
9:         $\widetilde{\boldsymbol{\Lambda}} = \widetilde{\boldsymbol{\Lambda}} \cup \boldsymbol{\Lambda}^l$
10:    **end for**
11:    $\lambda_\alpha = \min(\widetilde{\boldsymbol{\Lambda}}) + (\max(\widetilde{\boldsymbol{\Lambda}}) - \min(\widetilde{\boldsymbol{\Lambda}})) \times \alpha$
12:    $\mathcal{N} = \{\boldsymbol{\theta}^l | \lambda_{\max}(\boldsymbol{\Theta}_t^l) < \lambda_\alpha\}$ (*Modular policy Eq. 4*)
13:    $\mathcal{N} = \mathcal{N} \cup \{\boldsymbol{\theta}^l | \frac{|\Delta_t^l - \Delta_{t-1}^l|}{\Delta_1^l} < \beta\}$ (*Temporal policy Eq. 6*)
14:    $\mathcal{I} = \mathcal{M} - \mathcal{N}$
15:    **for** each batch $\mathbf{x}, \mathbf{y}$ in $\{\mathbf{x}_i, \mathbf{y}_i\}_{i=1}^n$ **do**
16:        **for** $\boldsymbol{\theta}^l$ in $\mathcal{I}$ **do**
17:            $\boldsymbol{\theta}^l = \boldsymbol{\theta}^l - \eta \nabla_{\boldsymbol{\theta}^l} \mathcal{L}(f(\mathbf{x}; \boldsymbol{\theta}), \mathbf{y})$
18:        **end for**
19:    **end for**
20: **end while**

---

2018; Bowman & Montufar, 2022). Empirical NTK, developed by many practical tools (Novak et al., 2022; Engel et al., 2022), succeeds in deep learning applications such as neural architecture search (Xu et al., 2021; Chen et al., 2021) and network pruning (Chen et al., 2023; Wang et al., 2023). Our work takes a deeper look into the modules of an over-parameterized model, demonstrating module-level training asynchronization of the components by the spatio-temporal distribution of the modular NTK. To the best of our knowledge, this is the first work that proposes to analyze the modular NTK variation for optimization.

**Adaptive Training Approaches.** Multirate training (Vlaar & Leimkuhler, 2022) is an emerging related technique that partitions neural network parameters into two parts, in which parameters in the slow part are updated less frequently. Our work analyzes the fine-grained training dynamics of modules and proposes a theoretical-inspired method to dynamically update parameters. Besides, there is a wide range of methods that can save computational resources, such as network pruning (Lee et al., 2018a; Rachwan et al., 2022) and dynamic sparse training (Liu et al., 2020; Jiang et al., 2022). These works tend to disable parameters during both the forward and backward propagation processes, while our work only sparsifies the gradients during the backward propagation process. Hence, these techniques are orthogonal to our method, and the proposed mNTK can be employed for evaluating the importance of parameters in pruning-based methods. We also integrate our method with pruning, and the results are presented in Appendix.

## 5 Experiments

This section presents experimental results on various model architectures. The models under consideration are both over-parameterized and highly structured , including BERT (Devlin et al., 2018)

with multi-head self-attention (MHSA), Switch-Transformer (Fedus et al., 2022) with both MHSA and mixture-of-experts (MoE) (Shazeer et al., 2017b) and VGG (Simonyan & Zisserman, 2014) with convolutional filters.

**Evaluation Measure.** We evaluate the effectiveness of our algorithm for model optimization in terms of *training efficiency* and *generalization*. The training efficiency is measured by the validation or test error of the models after a certain number of floating-point operations (FLOPs), which serves as a lower bound for the execution time (Justus et al., 2018). The generalization performance is measured by the converged test error. Additionally, we calculate the total number of FLOPs used by the models until they achieve the converged test error.

**Approximation of NTK.** It is hard to directly compute the empirical NTK for an over-parameterized model, due to the computational and memory cost when calculating the Jacobian matrix $J \in \mathbb{R}^{nk \times m}$. Therefore, two alternatives are widely used to approximate the empirical NTK: (1) sampling a subset of training data instead of using the entire set; (2) computing the gradient by sum-of-logits instead of all output units, as shown in (Mohamadi & Sutherland, 2022).

**Implementation Details.** In practice, we evaluate the empirical NTK by sampling N training data (N is dynamically adjusted based on the available GPU memory in the experimental setup). To expedite the NTK calculation, we leverage parallelism by utilizing the implementation of Engel et al. (2022)[2], which enables us to collect different data samples across multiple GPUs. The FLOPs is computed as the total sum of the forward and backward passes, where the backward FLOPs is approximately estimated as twice the forward pass (Rasley et al., 2020), and we measure it using the DeepSpeed Profiler[3]. All experiments are conducted on $8 \times$ NVIDIA GeForce RTX 3090 GPUs. For further experimental details, please refer to the Appendix.

## 5.1   Main Result

**Results of BERT.** Firstly, we experiment on BERT (Devlin et al., 2018), which stacks 12 Transformer layers with 12 attention heads in each layer. Following the basic setup of Liu et al. (2019), we train BERT from scratch by masked language modeling (MLM) task on WikiText-2 (Merity et al., 2016). We compare the proposed MAT method with directly training a BERT model (BERT), using multiple learning rates (Vlaar & Leimkuhler, 2022) to train BERT (Multirate), and randomly selecting some heads for training (BERT-Rand). In this experiment, MAT applies $\alpha = 0.1, \beta = 10^{-3}$.

Table 1: Results of BERT on WikiText-2. FLOPs is measured per GPU without embedding. Computation refers to the FLOPs model used until achieving best test loss. Best results are in boldface.

| Method | Validation Loss (Log PPL) | | | Test Loss (Log PPL) | Computation (PFLOPs) |
|---|---|---|---|---|---|
| | @ 10 PFLOPs | @ 15 PFLOPs | @ 20 PFLOPs | @ Convergence | |
| BERT | $5.39 \pm 0.15$ | $4.75 \pm 0.06$ | $4.48 \pm 0.03$ | $4.41 \pm 0.05$ | 28.70 |
| BERT-Rand | $5.65 \pm 0.18$ | $5.11 \pm 0.16$ | $4.96 \pm 0.11$ | $4.82 \pm 0.06$ | 21.53 |
| Multirate | $4.93 \pm 0.10$ | $4.57 \pm 0.03$ | $4.52 \pm 0.02$ | $4.48 \pm 0.03$ | 19.46 |
| MAT | $\mathbf{4.46 \pm 0.04}$ | $\mathbf{4.41 \pm 0.02}$ | $\mathbf{4.35 \pm 0.02}$ | $\mathbf{4.27 \pm 0.03}$ | $\mathbf{16.50}$ |

Table 1 shows the performance of all methods used in training BERT on WikiText-2. MAT outperforms all the baselines in training efficiency. In particular, MAT achieves the almost identical validation loss of 4.46 using 10 PFLOPs (1 PFLOPs = $10^{15}$ FLOPs) of computation whereas the vanilla BERT achieves 4.48 using 20 PFLOPs. Our method saves half of the computation by performing a lighter back propagation pass only for appropriate subsets of the modules. Compared to MAT, the performance degradation of BERT-Rand confirms the effectiveness of using $\lambda_{\max}$ as the criterion for selecting important modules. While Multirate shows the potential for efficient training, it comes at the cost of sacrificing performance. In contrast, MAT achieves better test performance with a reduction of about 15% in test perplexity, and also saves 41% of computational resources.

**Results of Switch-Transformer.** To further evaluate the scalability of our method, we apply MAT to the feed-forward network (FFN) of transformer layer. Switch-Transformer (Fedus et al., 2021) is a representative implementation of Mixture-of-Expert (MoE) architecture (Shazeer et al., 2017a),

---

[2]https://github.com/pnnl/torchntk
[3]https://github.com/microsoft/DeepSpeed

which has shown excellent performance in NLP tasks recently. MoE (Shazeer et al., 2017b) architects the FFN into a structured one, which replicates the FFN as experts and dynamically assigns tokens to each expert. In this experiment, we compare the performance of Switch-Transformers using vanilla, Multirate and Switch-Rand training methods on WikiText-103 (Merity et al., 2016). Specifically, MAT regards both experts and attention heads as the modules with $\alpha_h = 0.1, \alpha_e = 0.2, \beta_h = \beta_e = 10^{-3}$.

Table 2: Results of Switch-Transformer on WikiText-103. FLOPs is measured per GPU without embedding. Best results are in boldface.

| Method | Validation Loss (Log PPL) | | | Test Loss (Log PPL) | Computation (PFLOPs) |
|---|---|---|---|---|---|
| | @ 200 PFLOPs | @ 400 PFLOPs | @ 600 PFLOPs | @ Convergence | |
| Switch | $3.16 \pm 0.12$ | $2.31 \pm 0.05$ | $1.93 \pm 0.03$ | $1.74 \pm 0.02$ | 1155.2 |
| Switch-Rand | $2.92 \pm 0.15$ | $2.15 \pm 0.08$ | $2.01 \pm 0.07$ | $1.93 \pm 0.05$ | 837.5 |
| Multirate | $2.67 \pm 0.09$ | $2.02 \pm 0.08$ | $1.83 \pm 0.05$ | $1.77 \pm 0.04$ | 740.4 |
| MAT | $\mathbf{2.34 \pm 0.06}$ | $\mathbf{1.92 \pm 0.03}$ | $\mathbf{1.69 \pm 0.02}$ | $\mathbf{1.68 \pm 0.02}$ | 614.8 |

Table 2 shows that the performance of Switch-Transformer trained on a large dataset with a large number of parameters. It demonstrates that MAT significantly reduces the computation required for training. Switch-Rand and Multirate also improve training efficiency to some extent. Compared to them, our method precisely identifies the appropriate heads and experts trained with features that are easy to learn and generalize, reducing validation perplexity by 52.3%, 25.2%, and 18.1% (25.9%, 16.9% and 12.4% for log perplexity) within 200, 400, and 600 PFLOPs of computation, respectively. Compared to the vanilla model, our method achieves 10.4% improved test perplexity. These results demonstrate the effectiveness of MAT in larger models and datasets.

**Results of VGG.** To further validate the versatility of MAT, we deploy it on computer vision tasks. We take classic convolutional network VGG16 as an example, which is over-parameterized for the CIFAR-10 dataset. MAT splits convolutional filters into two parts of modules by $\alpha = 0.2, \beta = 10^{-6}$. Experimental results listed in Table 3 show that our method helps VGG16 converge faster (47.3%) with better test accuracy compared with the vanilla model.

Table 3: Results of VGG16 on CIFAR-10. Best results are in boldface.

| Method | Computation until Train Acc = n (PFLOPs) | | Test Accuracy (Top-1, %) | Computation (PFLOPs) |
|---|---|---|---|---|
| | @ n = 95% | @ n = 99% | @ Convergence | |
| VGG16 | 8.85 | 12.06 | $93.77 \pm 0.08$ | 17.14 |
| VGG16-Rand | 9.96 | 13.35 | $92.71 \pm 0.08$ | 18.84 |
| Multirate | 7.19 | 9.75 | $93.43 \pm 0.14$ | 13.35 |
| MAT | **5.31** | **7.21** | $\mathbf{93.86 \pm 0.05}$ | 9.03 |

## 5.2 Module-level Training Analysis

This experiment analyzes the histogram of the number of training epochs of all the attention heads in BERT. As shown in Figure 5, more than half of the heads were trained with backpropagation in only 20% of the epochs, and the average training epoch per head was 34% of the entire training process. This verifies that the sparse backpropagation induced by MAT enhances training efficiency. Furthermore, approximately 20% of the heads were never updated, indicating the sparse activation of over-parameterized structured models. This finding suggests the possibility of further reducing the computation of stable modules during the forward pass.

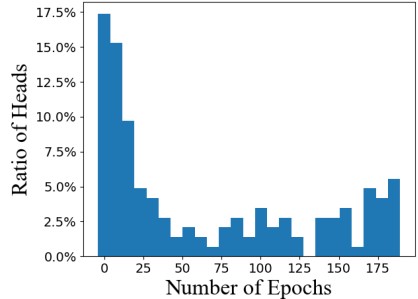

Figure 5: Histogram of epochs where the heads are trained using back-propagation.

# 6 Conclusion

We have analyzed the modular-level and temporal training characteristic of structured over-parameterized models through a new measure of modular neural tangent kernel (mNTK). Empirical and theoretical analysis demonstrates the relationship between optimization effectiveness and mNTK principal eigenvalues. Based on this finding, we designed a novel training strategy termed Modular Adaptive Training (MAT), which uses a modularly and temporally adaptive dynamic threshold to select partial modules for gradient back propagation. Experimental results show that MAT reduces training cost and increases test accuracy compared to existing training scheme. This work seeks to improve training efficiency by sparsifying the gradient update. Besides pruning, this work can be combined with other techniques to further improve training efficiency. We leave it for future work.

## Acknowledgements

This work was supported by National Natural Science Foundation of China under Grant No. 62090025, National Key R&D Program of China under Grant No. 2022YFB4400400 and China Postdoctoral Science Foundation No. 2022M720767.

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

# Appendix

## A  Trainability and Generalization

### A.1  Trainability

Following the previous work (Jacot et al., 2018) training neural networks in function space instead of parameter space, we analyze the trainability of an over-parameterized model by investigating the evolution of its predictions. Specifically, after a step of gradient descent

$$\boldsymbol{\theta}_{t+1} = \boldsymbol{\theta}_t - \eta \nabla_{\boldsymbol{\theta}_t} \mathcal{L}, \tag{7}$$

the updated predictions can be approximated using the first-order Taylor expansion as

$$
\begin{aligned}
f(\mathcal{X}; \boldsymbol{\theta}_{t+1}) &= f(\mathcal{X}; \boldsymbol{\theta}_t - \eta \nabla_{\boldsymbol{\theta}_t} \mathcal{L}) \\
&\approx f(\mathcal{X}; \boldsymbol{\theta}_t) - \eta \nabla_{\boldsymbol{\theta}_t} f(\mathcal{X}; \boldsymbol{\theta}_t)^\top \nabla_{\boldsymbol{\theta}_t} \mathcal{L} \\
&= f(\mathcal{X}; \boldsymbol{\theta}_t) - \eta \nabla_{\boldsymbol{\theta}_t} f(\mathcal{X}; \boldsymbol{\theta}_t)^\top \nabla_{\boldsymbol{\theta}_t} f(\mathcal{X}; \boldsymbol{\theta}_t) \nabla_{\mathcal{Z}} \mathcal{L} \\
&= f(\mathcal{X}; \boldsymbol{\theta}_t) - \eta \boldsymbol{\Theta}_t(\mathcal{X}, \mathcal{X}) \nabla_{\mathcal{Z}} \mathcal{L}.
\end{aligned}
\tag{8}
$$

Motivated by Arora et al. (2019), we rewrite the equation into the form of norm and eigenpairs:

$$
\begin{aligned}
\| f(\mathcal{X}; \boldsymbol{\theta}_{t+1}) - f(\mathcal{X}; \boldsymbol{\theta}_t) \|_2^2 &\approx \| \eta \boldsymbol{\Theta}_t(\mathcal{X}, \mathcal{X}) \nabla_{\mathcal{Z}} \mathcal{L} \|_2^2 \\
&= \eta^2 (\boldsymbol{\Theta}_t(\mathcal{X}, \mathcal{X}) \nabla_{\mathcal{Z}} \mathcal{L})^\top (\boldsymbol{\Theta}_t(\mathcal{X}, \mathcal{X}) \nabla_{\mathcal{Z}} \mathcal{L}) \\
&= \eta^2 \nabla_{\mathcal{Z}} \mathcal{L}^\top \boldsymbol{\Theta}_t(\mathcal{X}, \mathcal{X}) \boldsymbol{\Theta}_t(\mathcal{X}, \mathcal{X}) \nabla_{\mathcal{Z}} \mathcal{L} \\
&= \eta^2 \nabla_{\mathcal{Z}} \mathcal{L}^\top (\sum_i^{nk} \lambda_i \mathbf{u}_i \mathbf{u}_i{}^\top)(\sum_i^{nk} \lambda_i \mathbf{u}_i \mathbf{u}_i{}^\top) \nabla_{\mathcal{Z}} \mathcal{L} \\
&\overset{(i)}{=} \eta^2 \nabla_{\mathcal{Z}} \mathcal{L}^\top (\sum_i^{nk} \lambda_i^2 \mathbf{u}_i \mathbf{u}_i{}^\top \mathbf{u}_i \mathbf{u}_i{}^\top) \nabla_{\mathcal{Z}} \mathcal{L} \\
&\overset{(ii)}{=} \eta^2 \nabla_{\mathcal{Z}} \mathcal{L}^\top \sum_i^{nk} \lambda_i^2 \mathbf{u}_i \mathbf{u}_i{}^\top \nabla_{\mathcal{Z}} \mathcal{L} \\
&= \eta^2 \sum_i^{nk} \lambda_i^2 (\mathbf{u}_i{}^\top \nabla_{\mathcal{Z}} \mathcal{L})^\top \mathbf{u}_i{}^\top \nabla_{\mathcal{Z}} \mathcal{L} \\
&= \eta^2 \sum_i^{nk} \lambda_i^2 (\mathbf{u}_i{}^\top \nabla_{\mathcal{Z}} \mathcal{L})^2,
\end{aligned}
\tag{9}
$$

where $(i)$ follows $\mathbf{u}_i{}^\top \mathbf{u}_j = 0, \forall i \neq j$ and $(ii)$ follows $\mathbf{u}_i{}^\top \mathbf{u}_i = \|\mathbf{u}_i\|^2 = 1$. Then we can derive Eq. 9 into:

$$\| f(\mathcal{X}; \boldsymbol{\theta}_{t+1}) - f(\mathcal{X}; \boldsymbol{\theta}_t) \|_2 = \sqrt{\sum_{i=1}^{nk} (\eta \lambda_i \mathbf{u}_i{}^\top \nabla_{\mathcal{Z}} \mathcal{L})^2}. \tag{10}$$

As we can see, at every step of the gradient descent, the model learns the target function faster along the eigen-directions corresponding to the larger eigenvalues.

Further, for the loss function assumed by squared error loss, we characterize the loss reduction by the following directional derivative (Wang et al., 2020):

$$\Delta \mathcal{L}(\boldsymbol{\theta}) = \lim_{\epsilon \to 0} \frac{\mathcal{L}(\boldsymbol{\theta} + \epsilon \nabla_{\boldsymbol{\theta}} \mathcal{L}(\boldsymbol{\theta})) - \mathcal{L}(\boldsymbol{\theta})}{\epsilon}, \tag{11}$$

using Taylor's first-order approximation $\mathcal{L}(\boldsymbol{\theta} + \epsilon\nabla_{\boldsymbol{\theta}}\mathcal{L}(\boldsymbol{\theta})) \approx \mathcal{L}(\boldsymbol{\theta}) + \epsilon\nabla_{\boldsymbol{\theta}}\mathcal{L}(\boldsymbol{\theta})^{\top}\nabla_{\boldsymbol{\theta}}\mathcal{L}(\boldsymbol{\theta})$,

$$\Delta\mathcal{L}(\boldsymbol{\theta}) \approx \nabla_{\boldsymbol{\theta}}\mathcal{L}(\boldsymbol{\theta})^{\top}\nabla_{\boldsymbol{\theta}}\mathcal{L}(\boldsymbol{\theta}), \tag{12}$$

expanded by chain rule $\nabla_{\boldsymbol{\theta}}\mathcal{L}(\boldsymbol{\theta}) = \nabla_{\boldsymbol{\theta}}f(\boldsymbol{\theta})\nabla_{\mathcal{Z}}\mathcal{L}(\boldsymbol{\theta})$ into

$$\begin{aligned}
\Delta\mathcal{L}(\boldsymbol{\theta}) &= (\nabla_{\boldsymbol{\theta}}f(\boldsymbol{\theta})\nabla_{\mathcal{Z}}\mathcal{L}(\boldsymbol{\theta}))^{\top}\nabla_{\boldsymbol{\theta}}f(\boldsymbol{\theta})\nabla_{\mathcal{Z}}\mathcal{L}(\boldsymbol{\theta}) \\
&= \nabla_{\mathcal{Z}}\mathcal{L}(\boldsymbol{\theta})^{\top}(\nabla_{\boldsymbol{\theta}}f(\boldsymbol{\theta})^{\top}\nabla_{\boldsymbol{\theta}}f(\boldsymbol{\theta}))\nabla_{\mathcal{Z}}\mathcal{L}(\boldsymbol{\theta}),
\end{aligned} \tag{13}$$

then following Eq. 1 and matrix decomposition of mNTK,

$$\begin{aligned}
\Delta\mathcal{L}(\boldsymbol{\theta}) &= \nabla_{\mathcal{Z}}\mathcal{L}(\boldsymbol{\theta})^{\top}(\boldsymbol{\Theta})\nabla_{\mathcal{Z}}\mathcal{L}(\boldsymbol{\theta}) \\
&= \nabla_{\mathcal{Z}}\mathcal{L}(\boldsymbol{\theta})^{\top}(\sum_{l=1}^{L}\boldsymbol{\Theta}^{l})\nabla_{\mathcal{Z}}\mathcal{L}(\boldsymbol{\theta}) \\
&= \mathcal{Y}^{\top}(\sum_{l=1}^{L}\sum_{i=1}^{nk}\lambda_{i}^{l}\mathbf{u}_{i}^{l}\mathbf{u}_{i}^{l\top})\mathcal{Y} \\
&= \sum_{l=1}^{L}\sum_{i=1}^{nk}\lambda_{i}^{l}(\mathbf{u}_{i}^{l\top}\mathcal{Y})^{\top}(\mathbf{u}_{i}^{l\top}\mathcal{Y}) \\
&= \sum_{l=1}^{L}\sum_{i=1}^{nk}\lambda_{i}^{l}\left(\mathbf{u}_{i}^{l\top}\mathcal{Y}\right)^{2}.
\end{aligned} \tag{14}$$

The directional derivative of the loss function is closely related to the eigenspectrum of mNTKs. As for the latter projection item $\left(\mathbf{u}_{i}^{l\top}\mathcal{Y}\right)^{2}$, Arora et al. (2019) have studied the relationship between the projection norm and labels, and they demonstrate that true labels generate better alignment with top eigenvectors, especially for the maximum one. Therefore, we can assume that $\left(\mathbf{u}_{i}^{l\top}\mathcal{Y}\right)^{2} \propto \lambda_{i}^{l}$ in a real dataset.

## A.2 Generalization

We denote the set of $n$ training instances as $\mathcal{X} = (\mathbf{x}, \dots, \mathbf{x}_n)$ and the target set as $\mathcal{Y} = (\mathbf{y}_1, \dots, \mathbf{y}_n)^{\top}$. A two-layer neural network with the ReLU activation function can be written as

$$f_{\boldsymbol{\theta},\boldsymbol{a}}(\mathcal{X}) = \frac{1}{\sqrt{m}}\sum_{r=1}^{m}a_r\sigma\left(\boldsymbol{\theta}_r^{\top}\mathcal{X}\right) \tag{15}$$

where $\boldsymbol{\theta}_1, \boldsymbol{\theta}_2, \dots, \boldsymbol{\theta}_m$ are the weight vectors in the first layer, $a_1, a_2, \dots, a_m$ are weights in the second layer. For simplicity, we denote $\boldsymbol{\theta} = (\boldsymbol{\theta}_1, \dots, \boldsymbol{\theta}_m)$ and $\boldsymbol{a} = (a_1, \dots, a_m)^{\top}$. Following (Arora et al., 2019), we assume the network is over-parameterized and is trained by gradient descent on the quadratic loss over dataset $(\mathcal{X}, \mathcal{Y})$. We freeze the second layer $\boldsymbol{a}$ and optimize the first layer $\boldsymbol{\theta}$ through gradient descent. Let $\boldsymbol{\theta}(0)$ and $\boldsymbol{\theta}(t)$ denote the weights initialized from scratch and the weights after $t$ iterations, respectively.

Under these settings, according to the Lemma 5.3 of (Arora et al., 2019), the embedding function $f_{\boldsymbol{\theta},\boldsymbol{a}}$ learned from GD is in a restricted class of neural networks with weights close to initialization $\boldsymbol{\theta}(0)$. Therefore, the Rademacher complexity of the two-layer function can be bounded as follows:

**Lemma 2** *Given $R > 0$, with probability at least $1 - \delta$ over the random initialization $(\boldsymbol{\theta}(0), \boldsymbol{a})$, simultaneously for every $B > 0$, the following function class*

$$\begin{aligned}
\mathcal{F}_{R,B}^{\boldsymbol{\theta}(0),\boldsymbol{a}} = \{f_{\boldsymbol{\theta},\boldsymbol{a}} : \|\theta_r - \theta_r(0)\|_2 &\leq R \; (\forall r \in [m]), \\
\|\boldsymbol{\theta} - \boldsymbol{\theta}(0)\|_F &\leq B\}
\end{aligned} \tag{16}$$

*has empirical Rademacher complexity bounded as:*

$$\mathcal{R}_S\left(\mathcal{F}_{R,B}^{\boldsymbol{\theta}(0),\boldsymbol{a}}\right) = \frac{1}{n}\mathbb{E}_{\boldsymbol{\varepsilon}\in\{\pm1\}^n}\left[\sup_{f\in\mathcal{F}_{R,B}^{\boldsymbol{\theta}(0),\boldsymbol{a}}}\sum_{i=1}^{n}\varepsilon_i f\left(\mathbf{x}_i\right)\right]$$

$$\leq \frac{B}{\sqrt{2n}}\left(1+\left(\frac{2\log\frac{2}{\delta}}{m}\right)^{1/4}\right) + \frac{2R^2\sqrt{m}}{\kappa} + R\sqrt{2\log\frac{2}{\delta}}. \tag{17}$$

Lemma 2 indicates that the Rademacher complexity is proportional to the weight distance from its initialization. Specifically, a greater weight distance implies a more significant amount of Rademacher complexity and is thus associated with weaker generalization ability. Following those intuitions, we extend the connection between weight distance and Rademacher complexity, initially portrayed in a two-layer network, into more generalized deep neural networks.

For deep models, as mentioned in (Hoffer et al., 2017), the weight distance from its initialization grows logarithmically with the number of iterations, which can be described as

$$\|\boldsymbol{\theta}(t) - \boldsymbol{\theta}(0)\|_F \propto \log t. \tag{18}$$

Combining Lemma 2 and Eq. 18, we can discover that as training iterations increase, the model's Rademacher complexity also grows with its weights more deviated from initializations, which subsequently impairs the model's generalization ability. However, solutions remain. As trainability varies across various modules, we can prevent the significant growth of Rademacher complexity and thus retain a satisfying model generalization by ignoring the updates of those modules with undesirable mNTK eigenvalues.

# B   Experiments

## B.1   Experimental Settings

**BERT and Switch-Transformer**. We generally follow the settings of Liu et al. (2019) to train BERT and Switch-Transformer from scratch using self-supervised Masked Language Modeling (MLM). In detail, we uniformly sample approximately 15% input tokens to replace them with a mask, and the learning objective is to predict those masked tokens accurately with a cross-entropy loss.

All experiments are conducted on $8 \times$ NVIDIA GeForce RTX 3090 GPUs, and we list those key hyperparameters in Table 4.

Table 4: Hyperparameters configuration in BERT and Switch-Transformer

| Hyperparameter | BERT | Switch-Transformer |
|---|---|---|
| Number of layers | 12 | 12 |
| Attention heads | 12 | 12 |
| Hidden dimension size | 768 | 768 |
| Dropout | 0.1 | 0.1 |
| Attention dropout | 0.1 | 0.1 |
| Sequence length | 512 | 512 |
| Batch size | 8 | 8 |
| Warmup steps | 0 | 12k |
| Max steps | 12k | 300k |
| Weight decay | 0 | 0.01 |
| Peak learning rate | 2e-4 | 1e-4 |
| Learning rate decay | Linear | Cosine |
| Adam $[\epsilon, \beta_1, \beta_2]$ | [1e-6, 0, 0] | [1e-6, 0.9, 0.99] |
| Number of experts | | 4 |
| Capacity factor | | 1.5 |

The baseline methods consist of: 1) the vanilla model, 2) the Multirate (Vlaar & Leimkuhler, 2022) algorithm, and 3) the adaptive training with a random selection of the updated modules (abbreviated as Rand).

For BERT, Multirate randomly partitions the heads into fast and slow groups into a $50\%/50\%$ split, with fast heads updated every step of stepsize $\eta$ and slow ones updated every $k = 5$ step of stepsize $k\eta$.

BERT-Rand randomly selects and updates $50\%$ heads in every epoch. MAT first randomly samples $64$ data samples to approximate $64$ eigenvalues for each head-based mNTK. The modular policy and temporal policy are set with $\alpha = 0.1, \beta = 10^{-3}$. The adaptive training algorithm starts from the $5^{\text{th}}$ epoch and performs every 8 epochs.

For Switch-Transformer, Multirate considers each head/expert as a single module and partitions heads/experts into fast and slow sets at $50\%/50\%$ ratios. The fast heads and experts are updated every step with current stepsize $\eta$, and the slow ones are updated every $k = 5$ steps with stepsize $k\eta$. Switch-Rand randomly selects $50\%$ heads or experts for updating at the beginning of each epoch. MAT first randomly samples $64$ data samples and separately approximate $64$ eigenvalues for each head-based mNTK and expert-based mNTK. The modular policy and temporal policy are set with $\alpha_h = 0.1, \beta_h = 10^{-3}$ and $\alpha_e = 0.2, \beta_e = 10^{-3}$ for head and expert, respectively. The adaptive training algorithm performs every 2 epochs after warming up.

To avoid the case that no heads or experts are updated in any layer, we utilize a protecting strategy that maintains the gradient of at least one head or expert in each layer for all baselines and our model.

**VGG**. All baselines of VGG are initialized with Kaiming initialization (He et al., 2015) and are trained with SGD for 200 epochs with an initial learning rate of $0.1$ and batch size of $128$. We decay the learning rate by $0.1$ at $1/2$ and $3/4$ of the total number of epochs. Specifically, Multirate randomly partitions the filters into fast and slow parts by $50\%/50\%$ in each layer. VGG-Rand randomly selects $50\%$ filters to update filters in each epoch. MAT approximates eigenvalues for each filter-based mNTK using random $64$ data samples. The modular policy and temporal policy are set with $\alpha = 0.2, \beta = 10^{-6}$. All experiments are repeated by three runs and the final computation costs are calculated on average.

## B.2   MAT and Network Pruning

Network pruning (Frankle & Carbin, 2018; Sanh et al., 2020; Liu et al., 2021) applies various criteria to determine the importance of different components and prunes those that are most redundant to compress model size, which often results in slight performance drops. The proposed MAT distinguishes from network pruning in that MAT only sparsifies modules during backward propagation, while pruning methods eliminate the forward and backward propagations of pruned modules. Besides, MAT is the first work to employ the principal eigenvalue of mNTK as the module selection criterion.

Our empirical study reveals some modules that are never or rarely selected by the proposed adaptive training method (MAT), showing potential for being pruned to achieve further computation savings. However, due to the complete removal of modules, existing network pruning methods may negatively impact the model's performance and generalizations. Based on those observations, we have explored whether we can apply MAT to network pruning methods to accelerate the training process or to improve performance.

We employ a BERT model for this experiment, and we prune 50% attention heads according to the ranking of $\lambda_{\max}$ of head-based mNTKs at the $15^{\text{th}}$ epoch. Across the training session, we use MAT with $\alpha = 0.2$ to introduce sparsity in the backward pass, resulting in approximately 25% sparsity of weight gradients.

Table 5: Results of BERT on WikiText-2 by pruning methods.

| Method | Sparsity (Pruning Ratio) | | Test Loss (Log PPL) | Computation (PFLOPs) |
|---|---|---|---|---|
| | Forward Pass | Backward Pass | @ Convergence | |
| Vanilla | 0% | 0% | 4.41 | 27.80 |
| SNIP (50%) | 50% | 50% | 4.39 | 19.02 |
| SNIP (75%) | 75% | 75% | 4.70 | 15.22 |
| MAT | 50% | ~75% | **4.32** | **12.03** |

Table 5 compares the extended MAT, the vanilla BERT model, and SNIP (Lee et al., 2018b) in terms of forward and backward sparsity.

SNIP is a widely used pruning method that operates based on connection sensitivity, enabling sparsity in over-parameterized models. In our implementation, we apply SNIP in a modular manner by calculating the connection sensitivity of each module. As shown in the Table, SNIP achieves a 31.6% reduction in computation when pruning 50% of the attention heads without any performance degradation. However, when the pruning ratio (sparsity) is increased to 75%, SNIP fails to achieve comparable performance with the vanilla model. This suggests that a large ratio of sparsity can have a negative impact on model performance.

In contrast, using the criteria of MAT, we prune 50% of the attention heads while training the remaining ones by MAT. This approach leads to a further acceleration of computations by 56.7% while slightly improving the overall performance. This experiment serves as an example highlighting the potential of MAT in network pruning, showcasing the trade-off between computation savings and performance maintenance.

### B.3 Computational Complexity and Overhead of MAT

The proposed MAT can introduce additional computational and memory overheads as it involves the calculation of mNTKs and their principal eigenvalues. However, we demonstrate in this subsection that MAT only yields a negligible proportion of extra computations, and we also report the numeric overhead results from experiments to support this claim.

To clarify, we have employed two strategies to accelerate the computation of mNTKs significantly: (1) sampling a subset of size $S(\ll n)$ for the NTK approximation instead of using the entire set, and (2) computing the gradient using the sum-of-logits approach instead of considering all output units. With those strategies, we approximate the Jacobian matrix $J \in \mathbb{R}^{nk \times m}$ with the approximated Jacobian matrix $\widetilde{J} \in \mathbb{R}^{S \times m}$, and we only need to perform $S$ additional gradient propagations and concatenate the gradients together. mNTKs are then computed with the approximated Jacobian matrix, and we perform the eigen-decomposition to mNTKs to obtain the principal eigenvalue.

Apart from those techniques mentioned above, the module division strategy also accelerates the matrix multiplication of the Jacobian. Unlike the integral NTK, MAT applies a lightweight NTK estimation by modular NTK that significantly reduces the computation time required, and this estimation can be scalable to deeper structured networks. The complexity of computing integral NTK is $O(Sm^2)$; however, the overall time complexity reduces to $O(LS(m/L)^2) = O(Sm^2/L)$ in MAT, assuming we are computing $L$ mNTKs. As for the singular value decomposition (SVD), since $S \ll n \ll m$, its complexity, $O(LS^3)$, can be far lower than others. Table 6 illustrates the comparison of computational complexities, showcasing MAT's significant computational advantages for NTK approximation. In short, **the overhead produced by MAT is negligible**.

Table 6: Complexity comparison, where $n$ denotes the number of training data, $m$ the number of parameters, $k$ the output dimension, $L$ the number of components, and $S$ the sample number.

| Complexity | Full | Our Approximation |
|---|---|---|
| NTK computaion | $O(nkm^2)$ | $O(Sm^2/L)$ |
| SVD computaion | $O(n^3k^3)$ | $O(LS^3)$ |

We also measure the actual computation overhead by MAT. Following the experimental setting of Turc et al. (2019), we apply the proposed MAT to BERT models with different network scales, namely BERT-Mini (L=4, H=256), BERT-Small (L=4, H=512), BERT-Medium (L=8, H=512), BERT-Base (L=12, H=768), and BERT-Large (L=24, H=1024). Table 7 demonstrates the computational costs across varying multi-scale BERT models, and we can see that MAT can save 26.4%, 33.4%, 39.4%, 40.6%, and 50.9% computations for BERT-Mini, BERT-Small, BERT-Medium, BERT-Base, and BERT-Large, respectively. These observations indicate that applying MAT to larger models can better improve training efficiency. Notably, when compared to the overall computational costs, the overheads introduced by MAT are extremely small and can be arguably ignored.

Table 7: The computation (PFLOPs) required for training to convergence of models of different scale on WikiText-2.

| Model | BERT-Mini | BERT-Small | BERT-Meduim | BERT-Base | BERT-Large |
|---|---|---|---|---|---|
| Vanilla computation | 1.44 | 4.28 | 9.52 | 27.80 | 61.25 |
| MAT computation | 1.06 | 2.85 | 5.77 | 16.50 | 30.09 |
| MAT overhead | 0.01 | 0.04 | 0.08 | 0.24 | 0.45 |

