# OpenReview forum: "Train Faster, Perform Better: Modular Adaptive Training in Over-Parameterized Models"
_NeurIPS.cc/2023/Conference — NeurIPS 2023 poster_

### Official Review · Reviewer_1e7h · 2023-06-29

**Soundness:** 3 good
**Presentation:** 4 excellent
**Contribution:** 4 excellent
**Rating:** 7
**Confidence:** 4

**Summary:**

The authors introduce a novel modification of the neural tangent kernel (NTK) called modular NTK (mNTK).
It is essentially the decomposition of the NTK as a sum of tangent kernels for the disjoint modules
that make up the network. This allows for module-level analysis of training dynamics. They find that
principle eigenvalue of mNTKs tends to be orders of magnitude larger than the others and that the variation
of these principle eigenvalues is a synchronous across modules. Since the directions associated with
these principle eigenvalues dominates gradient updates during training, they can be utilized to selectively
update subsets of the modules to better allocate computational resources during training. The authors
further characterize modules with smaller max eigenvalues as being more likely to learn "nuisance" features.
They introduce a novel optimization schema called modular adaptive training (MAT) that stops back propagation
to modules whose principle eigenvalue falls below a threshold or if variation of the principle eigenvalue
falls below a threshold.
They demonstrate that MAT produces models with equivalent performance to vanilla trained model while requiring
fewer FLOPs during training. Furthermore, they show that MAT trained models tend to generalize better.


**Strengths:**

- Paper is well-written.
- Method is novel and theoretically well motivated.
- Supported by experiments.
- The method is very general and has the potential for high impact.


**Weaknesses:**


For the language models, the paper only looks at perplexity scores of the pretrained model for text. I'm not the most
up to date on this, but I think typically practitioners are more concerned about the performance of the model
when fine-tuned on down stream tasks. I think comparing performance on (vanilla) fine-tuned downstream tasks of MAT vs vanilla pretrained models would strengthen the results.

Tying a bit into the previous point, papers such as RoBERTa indicate that continuing to train pretrained models after
the train perplexity has mostly stopped decreasing can lead to significant boosts in performance when fine-tuned on
downstream tasks. If the low $\lambda_\textrm{max}$ heads mostly learn diverse features, then this might be negatively impacted by MAT pretraining. If they mostly learn noise, then there shouldn't be much negative impact on this. Success
of MAT in this setting (as indicated by increased performance on downstream tasks at a given computational cost) would
greater strengthen the method.


**Questions:**


- Line 221 mentions how the hyperparameter $\alpha$ is relatively stable throughout training. This doesn't really make sense to me because how could a constant not be stable throughout training. Is this a typo? I think you may have meant to say something like $\lambda_\alpha$ is stable throughout training.
- Have you explored using MAT for the process of fine-tuning a pretrained model? Typically, fine-tuning takes far less time than pretraining, but it can still take a while to converge if the target task has many examples.

**Limitations:**

Yes

---

> ### Author Rebuttal · Authors · 2023-08-09
>
> > `Q1` For the language models, the paper only looks at perplexity scores of the pretrained model for text. I'm not the most up to date on this, but I think typically practitioners are more concerned about the performance of the model when fine-tuned on down stream tasks. I think comparing performance on (vanilla) fine-tuned downstream tasks of MAT vs vanilla pretrained models would strengthen the results.
>
> `A1` Thanks for your suggestion. We compare performance of vanilla and MAT pre-trained model on the SST-2 (sentiment recognition) from GLUE Benchmark in the following table.
>
> | Method | MLM (perplexity) $\downarrow$ | SST-2 (accuracy) $\uparrow$ |
> |---|---:|---:|
> | vanilla | 4.41 | 84.36 |
> | MAT | 4.27 | 85.47 |
>
>
> > `Q2` Tying a bit into the previous point, papers such as RoBERTa indicate that continuing to train pretrained models after the train perplexity has mostly stopped decreasing can lead to significant boosts in performance when fine-tuned on downstream tasks. If the low $\lambda_{\max}$ heads mostly learn diverse features, then this might be negatively impacted by MAT pretraining. If they mostly learn noise, then there shouldn't be much negative impact on this. Success of MAT in this setting (as indicated by increased performance on downstream tasks at a given computational cost) would greater strengthen the method.
>
> `A2` Thanks for your comments. It's in line with what we are thinking. Many works demonstrate similar conclusions that continuing pre-training the model whose perplexity has converged boosts the downstream task performance. However, a major challenge in this process is how to distinguish learning diverse, useful features and learning noise. The proposed MAT shows potential for indicating which modules are learning the informative features that are common and likely to generalize.
>
> We've conducted related experiments. We pretrain BERT from scratch by MLM task on AGNews tilL the validation perplexity converges, and save the snapshot called $A$. After that, we obtain $B$ and $C$ by continuing to pretrain $A$ for 10 epochs with vanilla method and MAT seperately. We fine-tuned the three models on AGNews by classification task and the experimental results are shown in Table.
>
> | Accuracy | A (Full-Trained) | B (Continue-Trained) | C (Adaptive-Trained) |
> |---|---:|---:|---:|
> | Full Data | 74.73 | 75.02 | **75.54** |
> | Long-tail Data | 73.69 | 74.23 | **75.00** |
>
> > `Q3` Line 221 mentions how the hyperparameter $\alpha$ is relatively stable throughout training. This doesn't really make sense to me because how could a constant not be stable throughout training. Is this a typo? I think you may have meant to say something like $\lambda_\alpha$ is stable throughout training.
>
> `A3` Thanks for your question. The modular policy is motivated by the analysis of the eigen-spectrum distribution in Section 2.3. We find it exhibits two distinct regions in Figure 4 with a significant threshold, and we try to locate the threshold, which corresponds to the $\lambda_\alpha$. In fact, due to the magnitude variation of eigenvalues during the training process, $\lambda_\alpha$ should also be dynamic. However, we empirically observe the threshold is relatively stable in the normalized distribution as Equation 4, leading us to assign the normalized threshold $\alpha$ as a constant. Thus, the hyperparameter $\alpha$ keeps constant and the threshold $\lambda_\alpha$ is located dynamically according to the magnitude of overall eigenvalues.
>
> > `Q4` Have you explored using MAT for the process of fine-tuning a pretrained model? Typically, fine-tuning takes far less time than pretraining, but it can still take a while to converge if the target task has many examples.
>
> `A4` Thanks for your question. We do consider MAT as a potential method for fine-tuning or transfer learning, because parameters updating tends to be sparse or low-rank in these tasks. Following your suggestion, we conduct the experiments that fine-tune a pre-trained BERT(obtained from Huggingface) on SST-2 data. Experimental results show MAT achieves better performance with less computation.
>
> | Method | Accuracy | Computation (PFLOPs) |
> |---|---:|---:|
> | vanilla | 92.96 | 5.71 |
> | MAT | 93.41 | 3.35 |

---

> > ### Comment · Reviewer_1e7h · 2023-08-15
> >
> > Thanks for your response. My comments have been adequately addressed, so I'll bump up the score

---

> > > ### Author Response · Authors · 2023-08-15
> > > **Thanks for reading our rebuttal**
> > >
> > > Thanks again for your valuable comments, which are very helpful to us.

---

### Official Review · Reviewer_afqQ · 2023-07-05

**Soundness:** 3 good
**Presentation:** 4 excellent
**Contribution:** 3 good
**Rating:** 7
**Confidence:** 3

**Summary:**

This paper introduces modular adaptive training based on the largest eigenvalue of the NTK. In particular, based on the assumption that  the principal eigendirection matters the most for generalization, the authors propose to dynamically turn off gradient updating on certain modules if they have small principal eigenvalues. Experiments are conducted on NLP and CV tasks to show the superiority of MAT.

**Strengths:**

It is very important to reduce computation resources for large model training. This paper focuses on an important topic, and the approach is principled. The presentation is very organized and the delivery is clear.

**Weaknesses:**

See questions.

**Questions:**

1. In Figure 2(a), can we also see a picture showing the distribution of all eigenvalues? Now even the 16th eigenvalue seems pretty large, is there very small eigenvalue?
2. In definition 1, can you write down the size of each matrix? Like $J\in\mathbb{R}^{nk\times p}$.
3. In figure 3, I wonder if there is a phenomenon of benign overfitting in this BERT task. If you keep training beyond 100 epochs to make preplexity even smaller, will you see a double descent on validation?
4. In figure 2(a), $\lambda_1$ of the 12th layer is still large close to 100 epoch. If you keep training will that eigenvalue eventually decays to $0$? What is special about this layer?
5. Around line 220, you mentioned the computation cost of backprop is reduced. Say at some point my first layer is in the information modules but the last layer is in the nuisance modules. In this case, the computation cost is as large as all modules are active, is this right? Because you need to backprop through the later layers.
6. Can you give more intuition on eq(6)? If my $\lambda_1(\Theta_1)$ and $\lambda_1(\Theta_0)$ are small, but $\lambda_1(\Theta_t)$ and $\lambda_1(\Theta_{t-1})$ are both large, then this equation can still be satisfied. Do you want to stop gradient in this scenario?
7. In algorithm 1, how large did you use for sample number $S$? How accurate is this estimation on the eigenvalues?
8. When you compute FLOPs, do you include the computations used for eigendecomposition?
9. Can we see the downstream performance of the MAT-trained BERT on a downstream task?

---

> ### Author Rebuttal · Authors · 2023-08-09
>
> > `Q1` In Figure 2(a), can we also see a picture showing the distribution of all eigenvalues? Now even the 16th eigenvalue seems pretty large, is there very small eigenvalue?
>
> `A1` Thanks for your question. The eigenvalues can be refered to *Figure A* of PDF in General Response. The computation of NTK considers each output unit, so the magnitude of the eigenvalues is related to the specific task. The number of output units in BERT's MLM task is equal to the vocubulary size, which makes the magnitude of the smallest eigenvalue in BERT MLM task can be as large as $10^9$ ~ $10^{10}$. Therefore, we observe their relative or normalized value instead, as *Figure 4* for all attention heads.
>
> > `Q2` In definition 1, can you write down the size of each matrix?
>
> `A2` Sure. Suppose the parameter vector is partitioned into $L$ disjoint modules $\boldsymbol{\theta}=\{\boldsymbol{\theta}^1,\boldsymbol{\theta}^2,...,\boldsymbol{\theta}^L\}$, each of which contains $m^1,m^2,...,m^L$ parameters respectively. The size of each matrix is annotated below:
>
> $J_{\boldsymbol{\theta}^l}(\mathcal{X})\in\mathbb{R}^{nk\times m^l}$, $J_{\boldsymbol{\theta}_p}(\mathcal{X})\in\mathbb{R}^{nk\times 1}$, $\boldsymbol{\Theta}^l(\mathcal{X},\mathcal{X})\in\mathbb{R}^{nk\times nk}$,$\boldsymbol{\Theta}(\mathcal{X},\mathcal{X})\in\mathbb{R}^{nk\times nk}$
>
> > `Q3` In Figure 3, I wonder ... double descent on validation?
>
> `A3` Thanks for your question. In our experiment, the validation perplexity keeps increasing when training for an additional 1000 epochs without descent on validation. Thanks for pointing it out, and we will further study on it.
>
> > `Q4` In Figure 2(a), ... about this layer?
>
> `A4` Thanks for your question. The 12th layer is the last Transformer layer, closest to the output and classifier. In our experiment, $\lambda_{\max}$ of the last layer is still large even it's trained more(~500) epochs. A possible explanation is that, in the later training process, due to the noisy dataset, the training error is not necessarily reduced to zero, so the residual makes its $\lambda_{\max}$ keep stable. In comparison, we conduct an experiment on MLP with MNIST, whose training loss and $\lambda_{\max}$ of the last layer are both converged to small value. Thanks for pointing out this interesting phenomenon, we will do more experiments to further explain why it occurs.
>
> > `Q5` Around line 220, ... backprop through the later layers.
>
> `A5` Thanks for your comments. When using MAT, most gradient computation of fixed modules can be left out even when earliest layers need to be updated.
>
> Take a linear layer $y=Wx$ in an arbitrary model as an example, it goes through two computation steps during the standard backward process: $\frac{\partial L}{\partial W}=\frac{\partial L}{\partial y}x^{\top}$ and $\frac{\partial L}{\partial x}=W^{\top}\frac{\partial L}{\partial y}$. If we split $W$ into equal-sized modules $W=cat(W_1,W_2,...,W_n)$, we can sparsify the two steps into: $\frac{\partial L}{\partial W}=cat(0,...,\frac{\partial L}{\partial y_i}x^\top,...,0)$ and $\frac{\partial L}{\partial x}=\sum_{i}W_i^\top\frac{\partial L}{\partial y_i}$, where $W_i$ is the informative module. In other words, if $k$ modules in a layer are in nuisance space, we reduce the backward computation to $(n-k)/n$ of the origin.
>
> With multiple layers, as long as the latter layer produces non-zero $\frac{\partial L}{\partial x}$, i.e., by keeping at least one module in each layer active, the gradient can continue to propagate to the earlier layers. In our empirical study, we have not observed instances where later layers end their learning before the earlier ones, and thus the backpropagation is performed uninterruptedly.
>
> This example is congruent with the multi-head attention structure, and the associated gradient sparsification is proved to guarantee the convergence [A].
>
> > `Q6` Can you give more intuition on eq(6)? ... in this scenario?
>
> `A6` Thanks for your question. Our intuition is to stop the module whose $\lambda_1(\boldsymbol{\Theta})$ has small variation. Referring to Figure 3, near the time point of overfitting, certain modules begin to satisfy this condition. As for the situation you mentioned, where $\lambda\_1(\boldsymbol{\Theta}\_{n})$ is larger than the initial one $\lambda\_1(\boldsymbol{\Theta}\_{0})$, is avoided by recording the initial value $\lambda\_1(\boldsymbol{\Theta}\_{0})$ after warming up or at the time $\lambda\_1(\boldsymbol{\Theta})$ reaches its largest value.
>
> >  `Q7` In algorithm 1, how large did you use for sample number $S$ ? How accurate is this estimation on the eigenvalues?
>
> `A7` Thanks for your question. The sample number $S$ is set according to the learning task and GPU memory. In our experiment, $S=64$ for BERT/Switch-Transformer and $S=128$ for VGG-16. We demonstrate the mNTK $\lambda_{\max}$ distribution of approximate NTK and true empirical NTK in *Figure B* of PDF in General Response, which are nearly identical.
>
> > `Q8` When you compute FLOPs, do you include the computations used for eigendecomposition?
>
> `A8` Thanks for your question. We demonstrate that MAT only yields a small proportion (<1.5%) of extra computations. *Table 7* shows the computational costs of varying multi-scale BERT models, and we can see that the overheads introduced by MAT are negligible. The complete computational complexity analysis and numerical results can be found in *Appendix B.3*.
>
> > `Q9` Can we see the downstream performance of the MAT-trained BERT on a downstream task?
>
> `A9` Thanks for your question. We compare performance of vanilla and MAT pre-trained model on the SST-2 (sentiment recognition) from GLUE Benchmark in the following table.
>
> | Method | MLM (perplexity) $\downarrow$ | SST-2 (accuracy) $\uparrow$ |
> |---|---:|---:|
> | vanilla | 4.41 | 84.36 |
> | MAT | 4.27 | 85.47 |
>
> ---
> **References:**
>
> [A] Alistarh, Dan, et al. "The convergence of sparsified gradient methods." Advances in Neural Information Processing Systems 31 (2018).

---

> ### Comment · Area_Chair_Yngg · 2023-08-19
>
> Hi Reviewer afqQ,
>
> Since the discussion with the authors is closing soon, could you please go over the rebuttal and provide some feedback?
>
> Regards,
>
> AC

---

> ### Author Response · Authors · 2023-08-19
> **Further experimental results**
>
> Suggested by the reviewers, we have applied MAT to transfer learning problems to further evaluate the generalization ability of our method. We have conducted the experiments on both typical structured NLP(BERT-base) and CV(ResNet-32) models. The downstream tasks are text(SST-2, IMDb) and image(CIFAR10/100) classification. The experimental results demonstrate our method saves the computation(41%~51%) while improves the task performance. These results further show the potential of our method on transfer learning(or fine-tuning), as a practical learning method for modern machine learning.
>
> | Model | Method | Task | Accuracy | Computation (PFLOPs) |
> |---|---|---|---:|---:|
> | BERT-base | vanilla | SST-2 | 92.96 | 5.71 |
> | BERT-base | MAT | SST-2 | 93.41 | 3.35 |
> | BERT-base | vanilla | IMDb | 93.39 | 2.32 |
> | BERT-base | MAT | IMDb | 93.47 | 1.14 |
> | ResNet-32 | vanilla | CIFAR10 | 96.31 | 2.61 |
> | ResNet-32 | MAT | CIFAR10 | 96.48 | 1.37 |
> | ResNet-32 | vanilla | CIFAR100 | 83.14 | 2.75 |
> | ResNet-32 | MAT | CIFAR100 | 83.20 | 1.51 |
>
> Please let us know if you still have any concern about this paper and we will be more than happy to discuss with you before the discussion period ends.

---

> > ### Comment · Reviewer_afqQ · 2023-08-19
> >
> > Thanks for the answers from the authors. My questions are addressed and the downstream performance looks good, especially given the amount of computation cost saved. I'm happy to raise my score.

---

> > > ### Author Response · Authors · 2023-08-19
> > > **Thanks for reading our rebuttal**
> > >
> > > Thanks again for your valuable comments, which are very helpful to us.

---

### Official Review · Reviewer_pEnu · 2023-07-08

**Soundness:** 3 good
**Presentation:** 4 excellent
**Contribution:** 3 good
**Rating:** 6
**Confidence:** 2

**Summary:**

In this work the authors propose a method for analyzing modules of a neural network, in particular their utility in test-time generalization, via a modular neural tangent kernel (mNTK). The mNTK is eigenspectrum the NTK of the network modules, e.g. different attention heads. In this work it was found that the larger the largest eigenvalue for a module's mNTK, (I will call this lam-max-mNTK for convenience) the more that module contributes features which are useful for generalizability. For example when, during training, a neural network's lam-max-mNTKs plateauing on a low value is indicative that the network has overfit. The authors also demonstrate that the modules' lam-mat-mNTKs evolve asynchronously during training, indicating that optimization of certain modules is more important during different epochs during training. Using this intuition the authors propose modular adaptive training (MAT), which weights modules' learning rates during training in proportion to the lam-max-mNTK of that module. The authors find that this improves training, achieving a minimum loss faster (around 1-5% faster). These experiments, including the analysis of generalizability, were done on BERT, although MAT was also performed on VGG, where the computational savings were even greater.

**Strengths:**

The paper is clearly written and pleasant to read.

The proposed analysis is interesting and would be of general interest to the deep learning community.

The computational improvements are nice.

**Weaknesses:**

The mNTK isn't super novel, as it is very similar to spectral analysis work which as been performed before, although I do think it is sufficiently novel enough for to ML venues.

The performance improvements, while nice, aren't large enough (at least for BERT) that I would expect MAT to see a large deployment in the future considering implementation seems somewhat involved. Perhaps more work is needed to make the improvement more substantial.

The experimental results are somewhat limited. In particular its hard to conclude that the observations in Section 2.2 generalize to neural networks in general considering this was just applied to one network. Also I think concluding that two regions in Figure 4 correspond to "info space" and "nuisance space" seems like a bit of a large jump and again, its just for one network. The split is interesting, but can we really conclude that the small eigenvalues are nuisances?

**Questions:**

Do we have some more evidence that your interpretation of mNTK is correct? Is there some reason we are more interested in languange models? Do the VGG lam-max-mNTK results align with those you got for BERT?

---

> ### Author Rebuttal · Authors · 2023-08-09
>
> > `Q1` The mNTK isn't super novel, as it is very similar to spectral analysis work which has been performed before, although I do think it is sufficiently novel enough for to ML venues.
>
> `A1` Thanks for your comments. Indeed, the spectrum of NTK has been applied to two-layer networks to analyze the optimization of the overall network [A,B]. However, one key contribution of our work is that we observe significant inter-module variance during training, specifically, some modules already converge and fall into the nuisance space, while others are still in the information space. To the best of our knowledge, our work, for the first time, indicates that mNTK is a good metric to measure the inter-module training dynamics, and presents an mNTK-based adaptive training method to optimize the network training process.
>
> > `Q2` The performance improvements, while nice, aren't large enough (at least for BERT) that I would expect MAT to see a large deployment in the future considering implementation seems somewhat involved. Perhaps more work is needed to make the improvement more substantial.
>
> `A2` Thanks for your comments. To further verify the potential of MAT, we consider the following improvements:
>
> - Scalability: In Appendix Table 7, we present the computational costs of varying multi-scale BERT models. MAT can save 26.4%, 33.4%, 39.4%, 40.6%, and 50.9% computations for BERT-Mini, BERT-Small, BERT-Medium, BERT-Base, and BERT-Large (L=24, H=1024), respectively. These findings indicate that applying MAT to larger models can better improve training efficiency. In addition, larger datasets are more likely to contain more superfluous or noisy features, making more modules unnecessary to update.
> - Network Pruning: Our approach is not limited to sparsification on gradient backpropagation computation. Our empirical study reveals some modules that are never or rarely selected by the proposed adaptive training method (MAT), showing potential for being pruned to achieve further computation savings. In other words, $\lambda_{\max}$ of mNTK can be criteria of structured pruning. The experimental results can be found in Appendix B.2.
> - Training Stability: Large model pre-training is often unstable and requires careful tuning of the learning rate to maintain stable convergence. A common practice is to use a learning rate scheduler to gradually reduce the learning rate. MAT prompts the model to update in a consistent direction, which to some extent avoids the instability caused by noise, allowing the model to keep a large learning rate and achieve higher training efficiency. Experimental results are shown in following table.
>
> | Method | Valid PPL @ 10 PFLOPs | Test PPL @ Final | Computation (PFLOPs) |
> |---|---:|---:|---:|
> | MAT (w/ lr scheduler) | 4.46 | 4.27 | 16.50 |
> | MAT (w/o lr scheduler) | 4.37 | 4.30 | 14.75 |
>
> > `Q3` Also I think concluding that two regions in Figure 4 correspond to "info space" and "nuisance space" seems like a bit of a large jump and again, its just for one network. The split is interesting, but can we really conclude that the small eigenvalues are nuisances? Do we have some more evidence that your interpretation of mNTK is correct?
>
> `A3` Thanks for your questions. These two terms, information space and nuisance space, are originated from Oymak et al. [C], and the idea of splitting is followed by other works. For example, Li et al. proposes clean and corrupted information and proves that the clean residual is aligned with the top singular direction of the Jacobian matrix whereas label noise is aligned with the small singular directions [D]. Please refer to the **General Response** for detailed explanation.
>
> To further verify the discrepancy between the two spaces, we conduct an ablation study which performs adaptive training on modules lying in the nuisance space. The comparison is listed below, where further verify the modules entered nuisance space learn superfluous or noisy features.
>
> | Method | Valid PPL @ 10 PFLOPs | Valid PPL @ 15 PFLOPs | Test PPL @ Final | Computation (PFLOPs) |
> |---|---:|---:|---:|---:|
> | vanilla | 5.39 | 4.75 | 4.41 | 28.70 |
> | MAT (nuisance) | 5.78 | 5.08 | 4.63 | 23.79 |
> | MAT (information) | 4.46 | 4.41 | 4.27 | 16.50 |
>
>
> > `Q4` The experimental results are somewhat limited. In particular its hard to conclude that the observations in Section 2.2 generalize to neural networks in general considering this was just applied to one network. Do the VGG lam-max-mNTK results align with those you got for BERT?
>
> `A4` Thanks for your questions. Prior works have found that parameter updates in over-parameterized models are sparse and low-rank, which opens up the possibility of neural network pruning and sparse training. Following your suggestion, we conduct the experiments in VGG. The result of eigen-spectrum distribution is similarly splitted, as shown in *Figure C* of PDF in General Response.
>
> ---
> **References:**
>
> [A] Arora, Sanjeev, et al. "Fine-grained analysis of optimization and generalization for overparameterized two-layer neural networks." International Conference on Machine Learning. PMLR, 2019.
>
> [B] Xiao, Lechao, Jeffrey Pennington, and Samuel Schoenholz. "Disentangling trainability and generalization in deep neural networks." International Conference on Machine Learning. PMLR, 2020.
>
> [C] Oymak, Samet, et al. "Generalization guarantees for neural networks via harnessing the low-rank structure of the jacobian." arXiv preprint arXiv:1906.05392 (2019).
>
> [D] Li, Mingchen, Mahdi Soltanolkotabi, and Samet Oymak. "Gradient descent with early stopping is provably robust to label noise for overparameterized neural networks." International conference on artificial intelligence and statistics. PMLR, 2020.

---

> > ### Comment · Reviewer_pEnu · 2023-08-14
> >
> > Thanks for the response. These are some fair points, I think the computational speedup is nice. I'll bump a point or two.
> >
> > I am still not very certain with my score, however.

---

> > > ### Author Response · Authors · 2023-08-15
> > > **Thanks for reading our rebuttal**
> > >
> > > We are greatly encouraged by your response.
> > >
> > > In addition to the scalability, pruning, and stability benefits, we are currently applying MAT to transfer learning problems and further evaluating the generalization ability of the method. We will report back to you with the results as they become available. In the meantime, please let us know if you have any other questions about this work.
> > >
> > > Thank you so much.

---

> > > > ### Author Response · Authors · 2023-08-19
> > > > **Further experimental results**
> > > >
> > > > Suggested by the reviewers, we have applied MAT to transfer learning problems to further evaluate the generalization ability of our method. We have conducted the experiments on both typical structured NLP(BERT-base) and CV(ResNet-32) models. The downstream tasks are text(SST-2, IMDb) and image(CIFAR10/100) classification. The experimental results demonstrate our method saves the computation(41%~51%) while improves the task performance. These results further show the potential of our method on transfer learning(or fine-tuning), as a practical learning method for modern machine learning.
> > > >
> > > > | Model | Method | Task | Accuracy | Computation (PFLOPs) |
> > > > |---|---|---|---:|---:|
> > > > | BERT-base | vanilla | SST-2 | 92.96 | 5.71 |
> > > > | BERT-base | MAT | SST-2 | 93.41 | 3.35 |
> > > > | BERT-base | vanilla | IMDb | 93.39 | 2.32 |
> > > > | BERT-base | MAT | IMDb | 93.47 | 1.14 |
> > > > | ResNet-32 | vanilla | CIFAR10 | 96.31 | 2.61 |
> > > > | ResNet-32 | MAT | CIFAR10 | 96.48 | 1.37 |
> > > > | ResNet-32 | vanilla | CIFAR100 | 83.14 | 2.75 |
> > > > | ResNet-32 | MAT | CIFAR100 | 83.20 | 1.51 |
> > > >
> > > > Please let us know if you still have any concern about this paper and we will be more than happy to discuss with you before the discussion period ends.

---

### Official Review · Reviewer_TyiH · 2023-07-13

**Soundness:** 2 fair
**Presentation:** 3 good
**Contribution:** 2 fair
**Rating:** 4
**Confidence:** 4

**Summary:**

Inspired by the Neural Tangent Kernel literature, this paper analyzes the training dynamics of over-parameterized neural networks from the perspective of modularity. In particular, the paper proposes to decompose the Jacobians of the NTK into those of modules. Here, the concept of "module" is a bit ill-defined, referring to some coherent computation block. The paper proposes to analyze the NTK within each module and in particular look at the eigenspectrum. It shows that empirically the first eigenvalues dominate within each module and that those eigenvalues are very different across modules. This provides an avenue for focusing mostly on those modules with a high MNTK eigenvalue. Results on different architectures and datasets show noticeable speedups. Finally, a connection between optimizing only high eigenvalue parts of the NTK and generalization is cited and corresponding empirical results are provided in this direction as well.

**Strengths:**

- The paper reads well
- It's an interesting topic and potentially impactful in improving our understanding of training dynamics.

**Weaknesses:**

- The biggest limitation for me is that the method of computing the NTK and then computing its eigenvalues seems quite expensive. Speedup experiments are performed using flops, which bypasses the actual running time (which would have to include the algorithm itself), defeating the purpose if it takes longer than the speed gain.

- As presented, the algorithm mostly claims speedups by not updating some modules, but if we need to update one of the earliest layers we still need to perform most backpropagation computations. Furthermore, at best, this algorithms provides a 2x (or 1.5x depending on how you count) improvement since it basically only affects the backward step.

- I found the connection of eigenvalues of mNTKs and generalization a bit week both conceptually/theoretically and in terms of empirical results. It's enough to show promise, but the connection with the distance to initialization is only tangential and improvements in generalization are not necessarily directly linked to the NTK eigenspectrum in the experiments.

Minor suggestion: I would put % of heads instead of number of heads in figure 5 to make it more immediately parseable.

**Questions:**

- I'm a bit surprised at the lack of role of interactions between Jacobians or similar terms across modules in the entire derivation. Given how little constraints there are in what constitutes a module, I would've expected to see some effect. Could you explain this in more detail?
- As mentioned under weaknesses, I would really appreciate a deeper comment on the entire computation time when accounting for the algorithm itself.

**Limitations:**

- The added compute should've been mentioned more clearly in my opinion.

- No ethics review necessary.

---

> ### Author Rebuttal · Authors · 2023-08-09
>
> > `Q1` The biggest limitation ... the speed gain.
> > I would really appreciate a deeper comment on the entire computation time when accounting for the algorithm itself.
>
> `A1` Thanks for your comment. The proposed method MAT will introduce additional computational overheads as it involves calculation of empirical NTK and eigen-decomposition. However, numerical overhead results from *Table 7* demonstrate that MAT only requires a negligible proportion (1%~1.5%) of extra computations. The complete computational complexity analysis and numerical results can be found in *Appendix B.3*.
>
> To accelerate the computation, we employed two strategies to approximate the Jacobian matrix $J\in\mathbb{R}^{nk\times m}$ with $\tilde{J}\in\mathbb{R}^{S\times m}$, where we sample $S$ data and use the sum-of-logits approach [A] instead of considering all output units. Furthermore, MAT applies a lightwight NTK estimation by modular NTK instead of integral NTK, which reduces the complexity from $O(Sm^2)$ to $O(Sm^2/L)$, assuming we are computing $L$ mNTKs.
>
> FLOPs comparison gives the upper bound of the time savings, and the actual wall-clock time is limited by the hardware performance and optimization. Under our experimental conditions, the wall-clock time is reduced by 31.5% compared by vanilla model. Note that computional cost of the algorithm itself is also included in the statistics.
>
> | Method | Test PPL @ Final | Computation (PFLOPs) | Wall-Clock Time (s) |
> |---|---:|---:|---:|
> | vanilla | 4.41 | 28.70 | 8021 |
> | MAT | 4.27 | 16.50 | 5494 |
>
> > `Q2` As presented, the algorithm ... the backward step.
>
> `A2` Thanks for your comments. When using MAT, most gradient computation of fixed modules can be left out even when earliest layers need to be updated.
>
> Take a linear layer $y=Wx$ in an arbitrary model as an example, it goes through two computation steps during the standard backward process: the gradient of parameter weight W $\frac{\partial L}{\partial W}=\frac{\partial L}{\partial y}x^{\top}$ and the gradient of the input vector $\frac{\partial L}{\partial x}=W^{\top}\frac{\partial L}{\partial y}$, where $\frac{\partial L}{\partial x}$ will be used to compute $\frac{\partial L}{\partial W}$ of the front layer. If we split $W$ into equal-sized modules $W=cat(W_1,W_2,...,W_n)$, which satisfies $y=cat(y_1, y_2,...,y_n)=cat(W_1x,W_2x,...,W_nx)$, we can sparsify the two steps into: $\frac{\partial L}{\partial W}=cat(0,...,\frac{\partial L}{\partial y_i}x^\top,...,0)$ and $\frac{\partial L}{\partial x}=\sum_{i}W_i^\top\frac{\partial L}{\partial y_i}$, where $W_i$ is the informative module. In other words, if $k$ modules in a layer are in nuisance space, we reduce the backward computation to $(n-k)/n$ of the origin.
>
> With multiple layers, as long as the latter layer produces non-zero $\frac{\partial L}{\partial x}$, i.e., by keeping at least one module in each layer active, the gradient can continue to propagate to the earlier layers. In our empirical study, we have not observed instances where later layers end their learning before the earlier ones, and thus the backpropagation is performed uninterruptedly. In addition, once the shallow layers are all in nuisance space, we can stop the gradient computation in those layers altogether.
>
> This example is congruent with the multi-head attention structure, and the associated gradient sparsification is proved to guarantee the convergence [B].
>
> Furthermore, our approach is not limited to sparsification on gradient backpropagation computation. We can further save the forward computation if we consider mNTK $\lambda_{\max}$ as network pruning criteria. *Table 5* demonstrates the comparison of applying MAT as pruning method. The complete experimental results can be found in *Appendix B.2*.
>
> > `Q3` I found the connection ... eigenspectrum in the experiments.
>
> `A3` Thanks for your comments. Please refer to the **General Response**.
>
> > `Q4` Minor suggestion: I would put % of heads instead of number of heads in figure 5 to make it more immediately parseable.
>
> `A4` Thanks for your suggestion. We will improve this figure in the revised version.
>
> > `Q5` I'm a bit surprised at ... expected to see some effect.
>
> `A5` Thanks for your question. While it might seem counterintuitive, mNTKs are calculated independently across modules, and its sum of mNTKs equals to the integral NTK; the derivation is shown below. $\boldsymbol{\Theta}(\mathcal X, \mathcal X)=\sum_{l=1}^L \sum\_{\boldsymbol{\theta}_p \in \boldsymbol{\theta}^l}J\_{\boldsymbol{\theta}^l}(\mathcal X) J\_{\boldsymbol{\theta}^l}(\mathcal X)^{\top}=\sum\_{l=1}^L \boldsymbol{\Theta}^l(\mathcal X, \mathcal X)$. Different from MLP, modern neural networks are inherently structured. Nevertheless, modules are correlated for model optimization. As presented in the paper, we observe significant inter-module variation during training, specifically, some modules already converge and fall into the nuisance space, while others are still in the information space. Our work indicates that mNTK is a good metric to measure the inter-module training dynamics, and presents an mNTK-based adaptive training method to optimize network training process.
>
> In addition, the granularity of the modular division will have an impact on our adaptive strategy, the following table evaluates the impact of division granularities. As we can see, attention head is a suitable granularity.
>
> | Method | Valid PPL @ 10 PFLOPs | Test PPL @ Final | Computation (PFLOPs) |
> |---|---:|---:|---:|
> | MAT (layer) | 4.64 | 4.37 | 19.40 |
> | MAT (head) | 4.46 | 4.27 | 16.50 |
> | MAT (half-head) | 4.53 | 4.29 | 17.26 |
>
> ---
> **References:**
>
> [A] Mohamadi et al. "A fast, well-founded approximation to the empirical neural tangent kernel." International Conference on Machine Learning. PMLR, 2023.
>
> [B] Alistarh, Dan, et al. "The convergence of sparsified gradient methods." Advances in Neural Information Processing Systems 31 (2018).

---

> ### Comment · Area_Chair_Yngg · 2023-08-19
>
> Hi Reviewer TyiH,
>
> Since the discussion with the authors is closing soon, could you please go over the rebuttal and provide some feedback?
>
> Regards,
>
> AC

---

> ### Author Response · Authors · 2023-08-19
> **Further experimental results**
>
> Suggested by the reviewers, we have applied MAT to transfer learning problems to further evaluate the generalization ability of our method. We have conducted the experiments on both typical structured NLP(BERT-base) and CV(ResNet-32) models. The downstream tasks are text(SST-2, IMDb) and image(CIFAR10/100) classification. The experimental results demonstrate our method saves the computation(41%~51%) while improves the task performance. These results further show the potential of our method on transfer learning(or fine-tuning), as a practical learning method for modern machine learning.
>
> | Model | Method | Task | Accuracy | Computation (PFLOPs) |
> |---|---|---|---:|---:|
> | BERT-base | vanilla | SST-2 | 92.96 | 5.71 |
> | BERT-base | MAT | SST-2 | 93.41 | 3.35 |
> | BERT-base | vanilla | IMDb | 93.39 | 2.32 |
> | BERT-base | MAT | IMDb | 93.47 | 1.14 |
> | ResNet-32 | vanilla | CIFAR10 | 96.31 | 2.61 |
> | ResNet-32 | MAT | CIFAR10 | 96.48 | 1.37 |
> | ResNet-32 | vanilla | CIFAR100 | 83.14 | 2.75 |
> | ResNet-32 | MAT | CIFAR100 | 83.20 | 1.51 |
>
> Please let us know if you still have any concern about this paper and we will be more than happy to discuss with you before the discussion period ends.

---

### Author Rebuttal · Authors · 2023-08-09

# General Response

Thank you very much for reviewing our manuscript and providing detailed and constructive comments, which have been very helpful for us to improve the quality of our work. Please see our answers to address the comments of individual reviewers. Additionally, enclosed pdf with figures is used for `Q4` of reviewer `pEnu`, `Q1` and `Q7` of reviwer `afqQ`.

Here, we provide a general response to the common concerns of reviewers (especially `Q3` of reviewer `TyiH`, `Q1` and `Q6` of reviewer `pEnu`), in terms of the key contributions of our work, mNTK and its implications, the split of information and nuisance space, and their relationship to generalization.

Firstly, we would like to point out that analyzing the eigenvalues of NTK is equivalent to analyzing the singular values of the Jacobian matrix, specifically $\lambda_i (JJ^\top) = \sigma_i^2(J)$, where $\lambda$ denotes the eigenvalue and $\sigma$ denotes the singular value. Next, several prior works have analyzed the relationship between generalization and the singular values of the Jacobian matrix. Oymak et al. notice the low-rank structure of the Jacobian matrix, and the features that fall on the lower part of the Jacobian singular value spectrum are hard to generalize [A]. Li et al. consider clean and noisy labels, splits the residual accordingly as presented below, and proves that the clean residual is aligned with the top singular vectors whereas label noise is aligned with the small singular vectors [B].

$$
\underbrace{\boldsymbol y-f(\boldsymbol W_t)}\_{\text {corrupted residual }}=\underbrace{\tilde{\boldsymbol y}-f(\boldsymbol W_t)}\_{\text {clean residual }}+\underbrace{\boldsymbol y-\tilde{\boldsymbol y}}\_{\text {label corruption }}
$$

Intuitively speaking, mNTK measures the correlation of the gradient that different data samples produce on a certain module. The eigenspectrum of mNTK measures how frequent  data features exisit in the dataset. Features occur frequently are related to the large eigenvalues, while data-specific features, generally considered as noise, related to the small eigenvalues. As the dataset becomes larger, the small eigenvalues, corresponding to the data-specific noise, become even smaller. Therefore, the gap between large and small eigenvalues becomes more significant.

One key contribution of our work is that we observe significant inter-module variance during training, specifically, some modules already converge and fall into the nuisance space, while others are still in the information space. Therefore, we use mNTK $\lambda_{\max}$ as an indicator of modular training.

We further introduce Radmacher complexity as evaluation of generalization ability from Lemma 2 in Appendix A.2:

**Lemma 2** Given $R>0$, with probability at least $1-\delta$ over the random initialization $(\boldsymbol{\theta}(0), \boldsymbol{a})$, simultaneously for every $B>0$, the following function class

$$
\mathcal F_{R, B}^{\boldsymbol{\theta}(0), \boldsymbol{a}}=\\{f_{\boldsymbol{\theta}, \boldsymbol{a}}:\|\theta_r-\theta_r(0)\|_2 \leq R~(\forall r \in[m]), \\
\|\boldsymbol{\theta}-\boldsymbol{\theta}(0)\|_F \leq B\\}
$$

has empirical Rademacher complexity bounded as:

$$
\mathcal R_S(
\mathcal F_{R, B}^{\boldsymbol{\theta}(0), \boldsymbol{a}})=
\frac{1}{n} \mathbb E_{\boldsymbol{\varepsilon} \in\{ \pm 1\}^n}[\sup_{f \in
\mathcal F_{R, B}^{\boldsymbol{\theta}(0), \boldsymbol{a}}
} \sum_{i=1}^n \varepsilon_i f(\mathbf{x}_i)] \\
\leq \frac{B}{\sqrt{2 n}}(1+(\frac{2 \log \frac{2}{\delta}}{m})^{1 / 4})+\frac{2 R^2 \sqrt{m}}{\kappa}+R \sqrt{2 \log \frac{2}{\delta}}.
$$

Lemma 2 indicates that Rademacher complexity is proportional to the weight distance from its initialization, where $\|\boldsymbol{\theta}-\boldsymbol{\theta}(0)\|_F \leq B$. Parameters updating of those modules falling into the nuisance space slightly contribute to the loss reduction but increase the Rademacher complexity.

In summary, during model training, modules already fallen into the nuisance space (with low mNTK $\lambda_{\max}$) are prone to learn superfluous or even noise features, which increase the Rademacher complexity with weak trainability, and deteriorate the model's generalization ability.

To the best of our knowledge, this work indicates for the first time that mNTK is a good indicator of inter-module training dynamics, and presents an mNTK-based adaptive training method to optimize the network training process.

---
**References:**

[A] Oymak, Samet, et al. "Generalization guarantees for neural networks via harnessing the low-rank structure of the jacobian." arXiv preprint arXiv:1906.05392 (2019).

[B] Li, Mingchen, Mahdi Soltanolkotabi, and Samet Oymak. "Gradient descent with early stopping is provably robust to label noise for overparameterized neural networks." International conference on artificial intelligence and statistics. PMLR, 2020.

---

### Decision · Program_Chairs · 2023-09-21

**Decision:**

Accept (poster)

**Comment:**

The paper proposes to study the modular-level training dynamic of transformer models from a neural tangent kernel view. It shows an interesting connection between the mNTK's principal eigenvalue and the generalization ability of modular. Reviewers generally appreciate the contribution of this paper.